# COMPLETE AND LIPSCHITZ CONTINUOUS INVARIANTS OF GRAPHS UNDER GEOMETRIC ISOMORPHISM IN $\mathbb{R}^n$

## ABSTRACT

Euclidean graphs embedded in $\mathbb{R}^n$ with unordered vertices and straight-line edges represent important real objects such as molecules whose atoms are connected by chemical bonds. Many real objects preserve their properties under any rigid motion from the special Euclidean group $SE(n)$. Embedded graphs were previously distinguished under such rigid motion or geometric isomorphism in $\mathbb{R}^n$. Experimental noise motivates new Lipschitz continuous invariants so that perturbations of all vertices up to $\varepsilon$ change the invariants up to a constant multiple of $\varepsilon$ in a suitable metric, whose running time should polynomially depend on the number of unordered vertices. We developed new complete invariants that are stable under noise, form a natural hierarchy, and distinguish all chemically different graphs in the QM9 database of 130K+ molecules within a few hours on a modest desktop.

## 1 MOTIVATIONS, PROBLEM STATEMENT, AND NEW CONTRIBUTIONS

Many rigid structures from star constellations to molecules are represented by geometric graphs in a Euclidean space, see Bonchev (1991). More precisely, a *Euclidean graph* $G \subset \mathbb{R}^n$ is a finite set of $m$ unordered (unlabeled) vertices located at distinct points of $\mathbb{R}^n$ and connected by straight-line edges. Forgetting all edges of $G \subset \mathbb{R}^n$ gives us the *vertex set* $V(G) \subset \mathbb{R}^n$ of $m$ unordered points.

A Euclidean graph can be disconnected and can have vertices $v$ of any *degree* that is the number of edges whose endpoint is $v$. Loops and multiple edges (with the same endpoints) do not appear in Euclidean graphs because all edges are straight line segments and can also intersect in theory.

Graphs can be considered under any *equivalence* relation that should satisfy the axioms: 1) *reflexivity*: $G \sim G$, 2) *symmetry*: if $G \sim F$ then $F \sim G$, 3) *transitivity*: if $G \sim F$ and $F \sim H$ then $G \sim H$. In chemistry, the simplest equivalence of molecules is the chemical composition, which is insufficient in practice, e.g. Fig. 1 (right) shows *stereoisomers* that have the same chemical compositions and non-equivalent rigid shapes with different chemical properties, see Rieder et al. (2023).

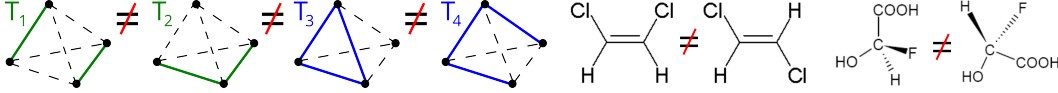

Figure 1: **Left**: graphs $T_i \subset \mathbb{R}^3$, $i = 1, 2, 3, 4$, on the same vertices with solid edges are not isomorphic to each other. **Right**: stereoisomers are isomorphic combinatorially, not geometrically.

For molecules, the strongest equivalence (distinguishing as many graphs as practically possible) is a *geometric isomorphism* $G \cong F$, i.e. an orientation-preserving transformation of $\mathbb{R}^n$ that bijectively maps the vertices and edges: $G \to F$. Geometric isomorphisms are also called *rigid motions* (compositions of translations and rotations), which form the special Euclidean group $SE(n)$.

Any geometrically isomorphic molecules have the same chemical properties. If a flexible molecule changes its rigid shape, its functional properties can change, so it is important to distinguish rigid shapes Wilson et al. (1991). The slightly weaker equivalence (not distinguishing mirror images) is *isometry*, which is any distance-preserving transformation including reflections. Since all real data (such as inter-point distances) are noisy, a more practically important answer is not binary ('same or different') but should be continuously quantified by a distance metric between isometry classes.

The ever-present atomic vibrations imply that rigid classes of molecules graphs on $m$ unordered atoms form a continuously infinite *Graph Isometry Space* $\text{GIS}(\mathbb{R}^3; m)$. Indeed, experimental noise and any iterative optimization can slightly perturb a molecular graph $G$ to a near-duplicate $F$ so that the rigid classes of $G, F$ are close to each other in $\text{GIS}(\mathbb{R}^3; m)$. The SSS theorem from school geometry implies that any triangular graphs are isometric if and only if they have the same triple of sides (inter-point distances) $a, b, c$ considered up to 6 permutations. Hence the space of triangular graphs is the triangular cone $\{0 < a \leq b \leq c \leq a + b\}$ in $\mathbb{R}^3$, where the last triangle inequality $c \leq a + b$ guarantees that three distances $a, b, c$ are realizable by a real triangle, see Fig. 2 (right).

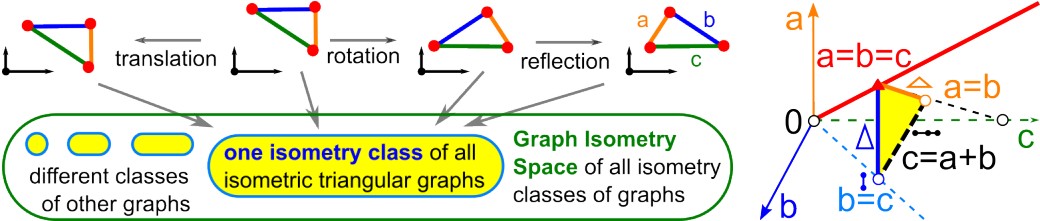

Figure 2: **Left**: the space of all isometry classes of graphs, each class has many representatives. **Right**: isometry classes of triangular graphs form the triangular cone $\{0 < a \leq b \leq c \leq a + b\}$ parametrized by inter-point distances $a, b, c$ with isosceles and degenerate triangles on the boundary.

To reliably distinguish any Euclidean graphs $G \subset \mathbb{R}^n$ similar to triangular graphs, we need an *invariant* $I$ defined as a numerical descriptor preserved by any rigid motion in $\mathbb{R}^n$. Alternatively, if $I(G) \neq I(F)$, then $G \ncong F$, so any invariant has *no false negatives* that are pairs of different representatives of *rigidly equivalent graphs* (denoted by $G \cong F$) having equal values of a (non-invariant) descriptor. The number of vertices (or edges) of $G$ is an integer-valued invariant that can distinguish some graphs but is too weak to separate the tetrahedral graphs $T_i$ in Fig. 1.

The strongest invariant $I$ separating all non-equivalent graphs is called *complete* meaning that if $I(G) = I(F)$ then $G \cong F$. Alternatively, a complete invariant $I$ has *no false positives* that are pairs of non-equivalent graphs $G \ncong F$ with $I(G) = I(F)$. Experimental noise in real data motivates continuous invariants that controllably change under perturbations, as formalized below.

**Problem 1.1** (complete invariant of Euclidean graphs with polynomial-time continuous metrics)**.** *For Euclidean graphs on $m$ unordered vertices in $\mathbb{R}^n$, find an invariant $I$ satisfying the conditions:*

*(a) completeness : any graphs $G, F$ are related by rigid motion in $\mathbb{R}^n$ if and only if $I(G) = I(F)$;*

*(b) Lipschitz continuity : there is a constant $\lambda$ and a metric $d$ on invariant values satisfying the axioms 1) $d(I, J) = 0$ if and only if $I = J$, 2) $d(I, J) = d(J, I)$, 3) $d(I, J) + d(J, K) \geq d(I, K)$, such that if $F$ is obtained by perturbing every vertex of $G$ up to $\varepsilon > 0$, then $d(I(G), I(F)) \leq \lambda\varepsilon$;*

*(c) invertibility : any graph $G \subset \mathbb{R}^n$ can be reconstructed (up to rigid motion in $\mathbb{R}^n$) from $I(G)$;*

*(d) computability : for a fixed dimension $n$, the invariant $I$, the metric $d$, and a reconstruction of $G \subset \mathbb{R}^n$ from $I(G)$ can be obtained in polynomial time of the number $m$ of unordered vertices.* ∎

Condition 1.1(a) means that a complete invariant $I$ is a DNA-style code that uniquely identifies any graph under geometric isomorphism. To be useful for noisy inputs, a complete invariant should continuously change under perturbations in a suitable metric. The axioms in 1.1(b) are the foundations of metric geometry Melter & Tomescu (1984) and accepted in physical chemistry Weinhold (1975). If the triangle axiom fails with any additive error, the classical $k$-means and DBSCAN clustering are open to adversarial attacks in Rass et al. (2024). If the first axiom is ignored, $d \equiv 0$ satisfies all other axioms. The first axiom implies the completeness of $I$ in 1.1(a) but the continuity is much stronger. Indeed, for any complete invariant $I$, one can define the discrete metric $d(I(G), I(F)) = 1$ for $G \ncong F$, which unhelpfully treats all non-equivalent graphs (even near-duplicates) as equally distant. The new requirement of Lipschitz continuity in 1.1(b) is much stronger than the classical $\varepsilon - \delta$ continuity because the constant $\lambda$ is universal for all $\varepsilon$ and unbounded graphs $G \subset \mathbb{R}^n$.

Condition 1.1(c) requires $I$ to be not only complete and continuous but also efficient to explicitly reconstruct $G$, better than a real DNA that doesn't explain how to grow a living organism. Computability 1.1(d) prevents brute-force attempts, e.g. defining $I(G)$ as the infinite set of images of $G$ under all rigid motions or taking $m!$ distance matrices over all permutations of $m$ unordered vertices.

**The main contribution** is the new invariant Nested Centered Distribution, which satisfies the conditions of Problem 1.1, including the new Lipschitz continuity, for all Euclidean graphs in any $\mathbb{R}^n$.

## 2 PAST WORK ON INVARIANTS AND DISTANCES FOR CLOUDS AND GRAPHS

This section reviews the related work starting from earlier partial solutions of Problem 1.1.

**Ordered clouds**. If a Euclidean graph $G \subset \mathbb{R}^n$ consists of $m$ isolated vertices without edges, the vertex set $V(G)$ can be called a *point cloud C*. If additionally, all points $p_1, \ldots, p_m$ of $C$ are ordered (not considered under the action of all $m!$ permutations), a complete invariant of $C$ under isometry (compositions of translations, rotations, reflections) is the classical $m \times m$ matrix Li et al. (2023) of pairwise distances $|p_i - p_j|$ due to (Grinberg & Olver, 2019, Theorem 9) or, after shifting the center of mass to the origin, the Gram matrix of scalar products $p_i Dot p_j$ by (Dekster & Wilker, 1987, Theorem 1). This multidimensional scaling known at least since 1935 Schoenberg (1935) can also provide an embedding $C \subset \mathbb{R}^k$ preserving all distances of $C$ for a dimension $k \leq m$. This embedding $C \subset \mathbb{R}^k$ uses eigenvectors whose ambiguity up to signs gives an exponential time that can be close to $O(2^m)$, not polynomial in the number $m$ of ordered points as required in 1.1(d).

**Unordered clouds**. Computational geometry developed many algorithms for detecting geometric isomorphism (or isometry, also called congruence) between point sets without edges Huttenlocher et al. (1993); Chew & Kedem (1992); Chew et al. (1999); Goodrich et al. (1999). (Arvind & Rattan, 2016, Theorem 3) describing, for a finite set $A \subset \mathbb{Q}^n$ of $m$ points, $n^{O(n)} poly(mM)$ time algorithm to compute a canonizing function $f(A)$, which can be considered a complete isometry invariant of $A$, where $M$ upper bounds the binary encodings of the rational coordinates in the input. For point clouds under rigid motion (also distinguishing mirror images), (Widdowson & Kurlin, 2023, Theorem 4.7) described a metric computable in time $O(n(m^{n-1}/n!)^3 \log m)$. The latest methods Hordan et al. (2024); Delle Rose et al. (2024); Amir et al. (2024); Maennel et al. (2024) also achieved the completeness for point clouds but without a Lipschitz continuous metric as in 1.1(b). Energy potentials written as infinite series of spherical harmonics, are often considered complete representations of atomic environments, which holds in the limit but not for a finite sizePozdnyakov et al. (2020).

For a fixed set of $m$ vertices in general position, one can choose any of $m(m-1)/2$ edges and produce $2^{m(m-1)/2}$ non-isometric graphs. Problem 1.1 for arbitrary graphs is computationally much harder than for point clouds due to exponentially many different graphs on the same vertex set.

**The graph isomorphism** problem Grohe & Schweitzer (2020) for abstract (non-Euclidean) graphs is another version of Problem 1.1 without continuous metrics. The latest advances Babai (2019); Helfgott et al. (2017) achieved only quasipolynomial time, though a polynomial time was again announced in Ohto (2024). While many partial cases were solved, e.g. for planar graphs (embedded in $\mathbb{R}^2$ without intersecting edges), see Kiefer et al. (2019), the $k$-dimensional Weisfeiler-Leman test Leman & Weisfeiler (1968) fails for 3-regular graphs of size $O(k)$. The key limitation of WL tests is their local nature when invariants are gradually expanded from a vertex or a $k$-tuple. Then covering a graph on $m$ vertices needs $O(m)$ expansions leading to exponential sizes in $m$. (Dym & Gortler, 2024, Section 3.9) discussed that a complete invariant (under all permutations of $m$ vertices) that has a polynomial time in the dimension $n$ would also solve the graph isomorphism problem in polynomial time. Condition 1.1(d) is easier for a fixed $n$ such as the practical dimensions $n = 2, 3$. The number $m$ of vertices can be dozens or hundreds, e.g. for molecular graphs in $\mathbb{R}^3$, where vertices are centers of atoms and edges are inter-atomic bonds that keep atoms together in a stable molecule.

**Geometric Deep Learning** in Bronstein et al. (2021) pioneered an axiomatic approach to geometric classifications beyond Euclidean space $\mathbb{R}^n$ in Bronstein et al. (2017). Many neural networks were proved to be universal Maron et al. (2019); Zhou (2020); Abbe & Sandon (2020) in the sense of approximating any continuous function on given data with sufficiently many layers. This universality property has been strengthened in Problem 1.1 to the full completeness of an explicit invariant that should be computable in polynomial time and invertible to an original graph up to rigid motion. The key challenge was to compute an exact (not approximate) metric that is also Lipschitz continuous.

**Equivariants** Kondor & Trivedi (2018); Cohen et al. (2019); Fuchs et al. (2020); Deng et al. (2021) are defined as descriptors $E$ satisfying $E(f(G)) = T_f(E(G))$ for any rigid motion $f$ of a graph $G$ in $\mathbb{R}^n$, where $T_f$ may not be the identity as required for invariants. Any linear combination of points, e.g. the center of mass, is equivariant but cannot distinguish graphs under translation.

Equivariants Gao et al. (2020); Qi & Luo (2020); Tu et al. (2022); Batzner et al. (2022) help predict forces acting on atoms to move them to a more optimal configuration. These time-dependent graphs $G_t$ can be studied directly by invariant values $I(G_t)$ without computing intermediate atomic forces. Many neural networks optimize millions of parameters as in (Goyal et al., 2021, Table 4) to improve accuracies Dong et al. (2018); Akhtar & Mian (2018); Laidlaw & Feizi (2019); Guo et al. (2019); Colbrook et al. (2022) but require re-training on any new data. All known descriptors of molecular graphs Duvenaud et al. (2015); Choo et al. (2023) have no proofs of all conditions 1.1(a,b,c,d).

**Gromov-Wasserstein** metrics Mémoli (2011) are defined for any metric-measure spaces Brécheteau (2019) by minimizing over infinitely many correspondences between points, but cannot be approximated with a factor less than 3 in polynomial time unless P=NP (Schmiedl, 2017, Corollary 3.8) and polynomial algorithms for partial cases in Mémoli et al. (2021); Majhi et al. (2024). The related problems of matching and finding distances between fixed Euclidean graphs (but not for their isometry classes) were studied in Nikolentzos et al. (2017); Majhi & Wenk (2022); Buchin et al. (2023). Computing a metric between rigid classes of clouds is only a small part of Problem 1.1. Indeed, to efficiently navigate on Earth, in addition to distances between cities, we need a map of the whole planet and hence an invertible continuous invariant $I$, which is an analog of geographic coordinates.

## 3 A HIERARCHY OF GRAPH INVARIANTS FROM FASTEST TO COMPLETE

Let $|p - q|$ denote the Euclidean distance between any points $p, q \in \mathbb{R}^n$. We always translate any graph $G \subset \mathbb{R}^n$ so that the *center of mass* $O(G) = \frac{1}{m} \sum_{p \in V(G)} p$ of the *vertex set* $V(G)$ is at the origin $0 \in \mathbb{R}^n$. Then Problem 1.1 reduces to the $\mathrm{SO}(n)$-invariance under orthogonal maps.

**Definition 3.1** (signed distance $d(p, q)$ and invariants $\mathrm{SRV}, \mathrm{SDV}, \mathrm{PDD}$). *Let $G \subset \mathbb{R}^n$ be any Euclidean graph on $m$ arbitrarily ordered vertices $p_1 \ldots, p_m$. If any $p_i, p_j \in V(G)$ are connected by an edge of $G$, define the* signed distance *as $d(p, q) = |p - q|$, else set $d(p, q) = -|p - q|$.*

*(a) The* Sorted Radial Vector $\mathrm{SRV}(G)$ *has $m$ distances $|p|$ for all $p \in V(G)$ in decreasing order.*

*(b) The* Sorted Distance Vector $\mathrm{SDV}(G)$ *has all $\frac{m(m-1)}{2}$ distances $d(p_i, p_j)$ in decreasing order.*

*(c) Let $D(G)$ be the $m \times (m - 1)$-matrix where the $i$-th row consists of signed distances $d(p_i, p_j)$, $j \in \{1, \ldots, m\} - \{i\}$ in increasing order. The* Pointwise Distance Distribution $\mathrm{PDD}(G)$ *is the distribution of unordered rows $D(G)$ with equal weights $1/m$.* ∎

If any $k > 1$ rows of $D(G)$ are equal, they can be collapsed in $\mathrm{PDD}(G)$ to a single row with the *weight $k/m$*. The PDD was defined for finite clouds as a local distribution of distances in (Mémoli, 2011, Definition 5.5) and for periodic sets in Widdowson & Kurlin (2022) but not for graphs.

Table 1: Acronyms and references of the main objects, invariants, and metrics in the paper.

| | | | | | |
|---|---|---|---|---|---|
| GIS | Graph Isometry Space | Fig. 2 | PDD | Pointwise Distance Distribution | Def 3.1 |
| SRV | Sorted Radial Vector | Def 3.1 | CD | Centered Distribution | Def 3.5 |
| SDV | Sorted Distance Vector | Def 3.1 | NCD | Nested Centered Distribution | Def 3.5 |
| CR | Centered Representation | Def 3.3 | NBM | Nested Bottleneck Metric | Def 4.5 |

The $\mathrm{PDD}(G)$ includes every signed distance twice, once as $d(p, q)$ in the row of a vertex $p$, and as $d(q, p)$ in the row of a vertex $q$. Hence $\mathrm{SDV}(G)$ can be obtained from $\mathrm{PDD}(G)$ by (1) combining all distances into one vector, (2) sorting them in decreasing order, and (3) keeping only one copy of every two repeated distances. Example 3.2 shows that $\mathrm{PDD}(G)$ is already stronger than $\mathrm{SDV}(G)$.

**Example 3.2** (invariants $\mathrm{SRV}, \mathrm{SDV}, \mathrm{PDD}$ for tetrahedral graphs in Fig. 1). *(a) Since the vertex sets of $T_i \subset \mathbb{R}^3$ are regular tetrahedra with all pairwise distances 1, these graphs have identical $\mathrm{SRV}(T_i)$ of 4 equal circumradii of the same vertex set $V(T_i)$ independent of $i = 1, \ldots, 4$.*

*The first graph $T_1$ has two edges contributing $+1$ and four non-edges (dashed lines) contributing $-1$ to the Sorted Distance Vector $\mathrm{SDV}(T_1) = (+1, +1, -1, -1, -1, -1)$. The graph $T_2$ also has*

*two edges, so* $\mathrm{SDV}(T_2) = \mathrm{SDV}(T_1)$ *doesn't distinguish* $T_1 \not\cong T_2$ *up to rigid motion. Similarly, the graphs* $T_3 \not\cong T_4$ *are not distinguished by* $\mathrm{SDV}(T_3) = (+1, +1, +1, -1, -1, -1) = \mathrm{SDV}(T_4)$.

**(b)** *In* $T_1$, *every vertex has exactly one edge and two non-edges (dashed lines), hence its signed distances are* $+1, -1, -1$. *The matrix* $\mathrm{PDD}(T_1) = (100\% \mid -1, -1, +1)$ *consists of a single row, where the weight* $100\%$ *indicates that all vertices of* $T_1$ *have the same row in* PDD. *The graph* $T_2$ *has one vertex (25%) with no edges, two vertices (50%) with one edge, and one vertex (25%) with two*

$$\text{edges, so } \mathrm{PDD}(T_2) = \left( \begin{array}{c|ccc} 25\% & -1 & -1 & -1 \\ 50\% & -1 & -1 & +1 \\ 25\% & -1 & +1 & +1 \end{array} \right) \neq \mathrm{PDD}(T_1), \text{ so } \mathrm{PDD} \text{ distinguishes the rigidly}$$

*non-equivalent graphs* $T_1 \not\cong T_2$ *with* $\mathrm{SDV}(T_1) = \mathrm{SDV}(T_2)$. *The graph* $T_3$ *has one vertex (25%)*

*with no edges and three vertices (75%) with two edges, so* $\mathrm{PDD}(T_3) = \left( \begin{array}{c|ccc} 25\% & -1 & -1 & -1 \\ 75\% & -1 & +1 & +1 \end{array} \right)$.

*The graph* $T_4$ *has two vertices (50%) with one edge and two vertices (50%) with two edges. Then*

$\mathrm{PDD}(T_4) = \left( \begin{array}{c|ccc} 50\% & -1 & -1 & +1 \\ 50\% & -1 & +1 & +1 \end{array} \right)$, *so* $\mathrm{PDD}$ *distinguishes the graphs* $T_3 \not\cong T_4$. ∎

For a graph $G$ with $m$ unordered vertices, $\mathrm{PDD}(G)$ has $m - 1$ columns. The reduced version $\mathrm{PDD}(G; k)$ includes only the first $k$ columns for $1 \leq k < m - 1$. Though PDDs have unordered rows, they can be continuously compared by Earth Mover's Distance Rubner et al. (2000).

Fig. S4 in Pozdnyakov et al. (2020) described infinitely many non-isometric pairs of clouds $C, C' \subset \mathbb{R}^3$ with $\mathrm{PDD}(C) = \mathrm{PDD}(C')$. These counter-examples inspired the stronger invariants for graphs below. Any $n$ vectors $p_1, \dots, p_n \in \mathbb{R}^n$ can be written as columns in the $n \times n$ matrix whose determinant has $\mathrm{sign}(p_1, \dots, p_n)$, which is $\pm 1$ or $0$ (if $p_1, \dots, p_n$ are linearly dependent).

**Definition 3.3** (Centered Representation $\mathrm{CR}(G; A)$ of a graph with $A \subset V(G)$). *Let* $G \subset \mathbb{R}^n$ *be a graph on* $m$ *unordered points with the center of mass* $O(G) = 0$. *For any* $1 \leq h \leq n$, *fix a base sequence* $A$ *of ordered vertices* $p_1, \dots, p_h \in V(G)$. *If* $h = n$, *let* $\mathrm{sign}(A)$ *be the sign of the* $n \times n$ *determinant on the vectors* $p_1, \dots, p_n$, *else* $\mathrm{sign}(A) = 0$. *Let* $D(A)$ *be the matrix of signed distances between the ordered points* $0 = p_0, p_1, \dots, p_h$. *The matrix* $R(G; A)$ *has* $m - h$ *unordered columns, one for each vertex* $q \in V(G) - A$, *consisting of* $h + 1$ *distances* $d(q, p_i)$ *for* $i = 0, \dots, h$, *where* $p_0 = 0$. *The* Centered Representation $\mathrm{CR}(G; A)$ *is the triple* $[\mathrm{sign}(A), D(A), R(G; A)]$. ∎

**Example 3.4** (CRs for 2-point bases in $\mathbb{R}^2$). *Let* $G \subset \mathbb{R}^2$ *be the triangular cycle on* $p_1 = (2, 0)$, $p_2 = (-1, 1)$, $p_3 = (-1, -1)$, *so* $O(G) = 0$ *and all signed distances are positive, see Fig. 3 (top left). For* $A = (p_1, p_2)$, $\mathrm{sign}(A) = \mathrm{sign} \begin{vmatrix} 2 & -1 \\ 0 & 1 \end{vmatrix} = 1$. *The distance matrix*

*on* $0, p_1, p_2$ *is* $D(p_1, p_2) = \begin{pmatrix} 0 & 2 & \sqrt{2} \\ 2 & 0 & \sqrt{10} \\ \sqrt{2} & \sqrt{10} & 0 \end{pmatrix}$. *Then* $R(G; p_1, p_2) = \begin{pmatrix} |p_3| \\ |p_3 - p_1| \\ |p_3 - p_2| \end{pmatrix} = $

$\begin{pmatrix} \sqrt{2} \\ \sqrt{10} \\ 2 \end{pmatrix}$. *Then* $\mathrm{CR}(G; p_1, p_2) = [+1, D(p_1, p_2), R(G; p_1, p_2)]$. *Replacing* $p_2$ *with* $p_3$, *we find*

$\mathrm{sign}(p_1, p_3) = \mathrm{sign} \begin{vmatrix} 2 & -1 \\ 0 & -1 \end{vmatrix} = -1$, $D(p_1, p_3) = \begin{pmatrix} 0 & 2 & \sqrt{2} \\ 2 & 0 & \sqrt{10} \\ \sqrt{2} & \sqrt{10} & 0 \end{pmatrix}$, *and* $R(G; p_1, p_3) = $

$\begin{pmatrix} |p_2| \\ |p_2 - p_1| \\ |p_2 - p_3| \end{pmatrix} = \begin{pmatrix} \sqrt{2} \\ \sqrt{10} \\ 2 \end{pmatrix}$. *Then* $\mathrm{CR}(G; p_1, p_3) = [-1, D(p_1, p_3), R(G; p_1, p_3)]$. ∎

**Definition 3.5** (Nested Centered Distribution $\mathrm{NCD}(G; h)$ of order $h$). *Let* $G \subset \mathbb{R}^n$ *be any Euclidean graph on* $m$ *unordered vertices and the center of mass at the origin* $0 \in \mathbb{R}^n$. *Fix an order* $1 \leq h \leq n$. *For any ordered vertices* $p_1, \dots, p_{h-1} \in V(G)$, *the* Centered Distribution $\mathrm{CD}_{h-1}(G; p_1, \dots, p_{h-1})$ *is the unordered set of* $\mathrm{CR}(G; p_1, \dots, p_h)$ *for all* $p_h \in V(G) - \{p_1, \dots, p_{h-1}\}$. *For any* $1 < k < h$ *and* $p_1, \dots, p_{k-1} \in V(G)$, $\mathrm{CD}_{k-1}(G; p_1, \dots, p_{k-1})$ *is the unordered set of* $\mathrm{CD}_k(C; p_1, \dots, p_k)$ *for all* $p_k \in V(G) - \{p_1, \dots, p_{k-1}\}$. *The* Nested Centered Distribution $\mathrm{NCD}(G; h)$ *is the unordered set of* $\mathrm{CD}_1(G; p_1)$ *for all vertices* $p_1 \in V(G)$. *For the order* $h = n$, *define the* mirror image $\overline{\mathrm{NCD}}(G; n)$ *as* $\mathrm{NCD}(G; n)$ *after reversing* $\mathrm{sign}(p_1, \dots, p_n)$ *of* $n \times n$ *determinants in all* CR*s*. ∎

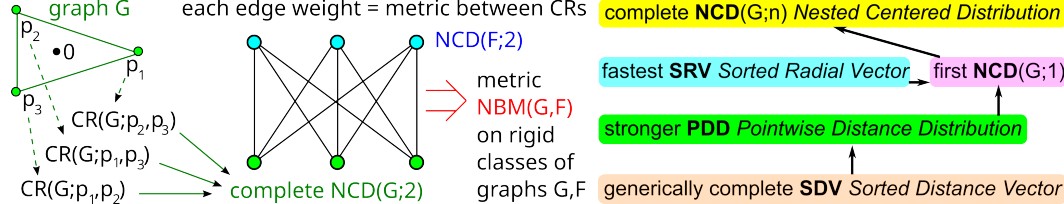

Figure 3: **Left**: building the Nested Centered Distribution NCD from Definition 3.5 from Centered Representations in Definition 3.3 with metrics in section 4. **Right**: hierarchy of graph invariants.

**Example 3.6.** *(a) For $h = 0$, the Nested Centered Distribution $\mathrm{NCD}(G; 0)$ consists of radial distances $|p|$ for all vertices $p \in V(G)$ and, after sorting in decreasing order, becomes $\mathrm{SRV}(G)$.*

*(b) For $h = 1$, $\mathrm{NCD}(G; 1)$ is the unordered collection of $\mathrm{CR}(G; p)$ for all vertices $p \in V(G)$. Here each Centered Representation $\mathrm{CR}(G; p)$ is a triple $[\mathrm{sign}(p), D(p), R(G; p)]$, where $\mathrm{sign}(p)$ is the usual sign of $p \in \mathbb{R}$ for dimension $n = 1$, otherwise $\mathrm{sign}(p) = 0$. The matrix of signed distances $D(p)$ consists of one signed distance $d(p, 0)$ between $p$ and the origin $0 \in \mathbb{R}^n$ (not considered as a vertex of G), so $D(p) = -|p|$. The $2 \times (m - 1)$ matrix $R(G; p)$ has unordered columns, one for each vertex $q \in V(G) - \{p\}$, consisting of the signed distance $d(0, q) = -|q|$ from 0 to q, and $d(q, p) = \pm|q - p|$, where we take $+$ if $q, p$ are joined by an edge of G. So $\mathrm{NCD}(G; 1)$ contains $\mathrm{NCD}(G; 0)$. All second rows of the matrices $R(G; p)$ with equal weights and without signs form the Pointwise Distance Distribution $\mathrm{PDD}(G)$ from Definition 3.1, see the hierarchy in Fig. 3 (right).* ∎

## 4 LIPSCHITZ CONTINUOUS METRICS ON THE NEW INVARIANTS OF GRAPHS

When a base sequence $A = (p_1, \ldots, p_n) \subset \mathbb{R}^n$ degenerates to a lower dimensional subspace, $\mathrm{sign}(A)$ of discontinuously changes. To guarantee the Lipschitz continuity, we multiply these signs by the strength $\sigma$ below, while the volume $\mathrm{vol}(A)$ of the simplex on $A$ is not Lipschitz continuous.

**Definition 4.1** *(strength $\sigma(A)$)*. *For any base sequence $A$ of $n$ ordered points $p_1, \ldots, p_n \in \mathbb{R}^n$, let $p(A) = \dfrac{1}{2} \sum\limits_{1 \leq i < j \leq n} |p_i - p_j|$ be a half-sum of all distances between the points of A. Let $\mathrm{vol}(A)$ denote the volume of the $n$-dimensional simplex on A. The strength is $\sigma(A) = \dfrac{\mathrm{vol}^2(A)}{p^{2n-1}(A)}$.* ∎

If $n = 1$, $A = \{p_0, p_1\} \subset \mathbb{R}$, then $\mathrm{vol}(A) = |p_1 - p_0| = 2p(A)$, so $\sigma(A) = \dfrac{\mathrm{vol}^2(A)}{p(A)} = 2|p_1 - p_0|$. If $p_0 = 0$, then $s(p_1) = 2p_1$. For $n = 2$ and a triangle $A$ with sides $a, b, c$, Heron's formula gives $\sigma(A) = \dfrac{(p-a)(p-b)(p-c)}{p^2}$. Here $p = \dfrac{a + b + c}{2} = p(A)$ is the half-perimeter of $A$.

**Lemma 4.2** *(Lipschitz continuity of $\sigma$, proved in (Widdowson & Kurlin, 2023, Thm 4.4))*. *Let $B$ be obtained from a sequence $A \subset \mathbb{R}^n$ of $n$ points by perturbing every point within its $\varepsilon$-neighborhood. Then $|\sigma(A) - \sigma(B)| \leq 2\varepsilon\lambda_n$ for a constant $\lambda_n$, where $\lambda_1 = 2$, $\lambda_2 = 2\sqrt{3}$, $\lambda_3 \approx 0.43$.* ∎

For any $k \times k$ matrices $M, N$ of real numbers, the metric $L_\infty$ is $\max\limits_{i,j=1,\ldots,k} |M_{ij} - N_{ij}|$. The *bottleneck* distance between any clouds $A, B$ of (the same number of) $m$ unordered points in a metric space with a metric $d$ is $W_\infty(A, B) = \min\limits_{g:A \to B} \max\limits_{p \in A} d(g(p), p)$ minimized for all bijections $g : A \to B$.

**Definition 4.3** *(max metric $M_\infty$ on CRs)*. *Let Euclidean graphs $G, F \subset \mathbb{R}^n$ on $m$ unordered vertices have base sequences $A, B$ of $h \leq n$ vertices. Consider the $m - h$ columns of $R(G; A)$ as a cloud of $m - h$ unordered points in $\mathbb{R}^h$, also for $R(F; B)$. The max metric $M_\infty(\mathrm{CR}(G; A), \mathrm{CR}(F; B))$ is the maximum of $\dfrac{2}{\lambda_n}|\mathrm{sign}(A)\sigma(0 \cup A) - \mathrm{sign}(0 \cup B)\sigma(B)|$, $L_\infty(D(A), D(B))$, and $W_\infty(R(G; A), R(F; B))$, where all signs are zeros for $h < n$.* ∎

To get a metric on Nested Centered Distributions, we will iteratively use the distance on bipartite graphs whose edge weights are the max metrics $M_\infty$ above on Centered Representations.

**Definition 4.4** (Bottleneck Matching Distance $\mathrm{BMD}(\Gamma)$). *Let $\Gamma$ be a complete bipartite graph with $m$ white vertices and $m$ black vertices so that every white vertex is connected to every black vertex by a single edge $e$ of a weight $w(e) \geq 0$. A* vertex matching *of the graph $\Gamma$ is a collection $E$ of $m$ disjoint edges. The* weight $W(E) = \max\limits_{e \in E} w(e)$ *is the largest weight of an edge in $E$. The* Bottleneck Matching Distance $\mathrm{BMD}(\Gamma) = \min\limits_{E} W(E)$ *is the minimum weight of a vertex matching $E$ of $\Gamma$.* ∎

**Definition 4.5** (Nested Bottleneck Metric NBM on NCDs). *Let $G, F \subset \mathbb{R}^n$ be any Euclidean graphs on $m$ unordered vertices. For any ordered vertices $p_1 \ldots, p_{h-1} \in V(G)$ and $q_1 \ldots, q_{h-1} \in V(F)$, the complete bipartite graph $\Gamma(G; p_1, \ldots, p_{h-1}; F; q_1, \ldots, q_{h-1})$ has $m - h + 1$ white vertices and $m - h + 1$ black vertices representing $\mathrm{CR}(G; p_1, \ldots, p_h)$ and $\mathrm{CR}(F; q_1, \ldots, q_h)$ for all $m - h + 1$ vertices $p_h \in V(G) - \{p_1, \ldots, p_{h-1}\}$ and $q_h \in V(F) - \{q_1, \ldots, q_{h-1}\}$, respectively.*

*Set the* weight $w(e)$ *of an edge $e$ joining the vertices represented by $\mathrm{CR}(G; p_1, \ldots, p_h)$, $\mathrm{CR}(F; q_1, \ldots, q_h)$ as the max metric $M_\infty$ between these distributions, see Definition 4.3. Then Definition 4.4 gives the bottleneck matching distance $\mathrm{BMD}(\Gamma(G; p_1, \ldots, p_{h-1}; F; q_1, \ldots, q_{h-1}))$.*

*For any integer $1 \leq i < h$ and ordered vertices $p_1 \ldots, p_{i-1} \in V(G)$ and $q_1 \ldots, q_{i-1} \in V(F)$, the complete bipartite graph $\Gamma(G; p_1, \ldots, p_{i-1}; F; q_1, \ldots, q_{i-1})$ has $m - i + 1$ white vertices and $m - i + 1$ black vertices representing $\mathrm{CD}_i(G; p_1, \ldots, p_i)$ and $\mathrm{CD}_i(F; q_1, \ldots, q_i)$ for all $m - i + 1$ variable vertices $p_i \in V(G) - \{p_1, \ldots, p_{i-1}\}$ and $q_i \in V(F) - \{q_1, \ldots, q_{i-1}\}$, respectively.*

*Set the* weight $w(e)$ *of an edge $e$ joining the vertices represented by $\mathrm{CD}_i(G; p_1, \ldots, p_i)$ and $\mathrm{CD}_i(F; q_1, \ldots, q_i)$ as the previously computed distance $\mathrm{BMD}(\Gamma(G; p_1, \ldots, p_i; F; q_1, \ldots, q_i))$ for a smaller number $i$ of fixed vertices. Then Definition 4.4 gives the bottleneck matching distance $\mathrm{BMD}(\Gamma(G; p_1, \ldots, p_{i-1}; F; q_1, \ldots, q_{i-1}))$. For $i = 1$, the graph $\Gamma(G, F)$ has $m + m$ vertices representing $\mathrm{CD}_1(G; p_1)$, $\mathrm{CD}_1(F; q_1)$ for all $p_1 \in V(G)$ and $q_1 \in V(F)$. The* Nested Bottleneck Metric $\mathrm{NBM}(\mathrm{NCD}(G; h), \mathrm{NCD}(F; h))$ *is the Bottleneck Matching Distance $\mathrm{BMD}(\Gamma(G, F))$.* ∎

**Example 4.6** (NBM for $h = 0, 1$). *Let $G \subset \mathbb{R}^n$ be any Euclidean graph on $m$ unordered vertices.*

*(a) For order $h = 0$, $\mathrm{NCD}(G; 0)$ after sorting in decreasing order coincides with $\mathrm{SRV}(G)$. Then the Nested Bottleneck Metric NBM coincides with the metric $L_\infty(\mathrm{SRV}(G), \mathrm{SRV}(F))$.*

*(b) For order $h = 1$, the invariant $\mathrm{NCD}(G; 1)$ is the unordered distribution of $\mathrm{CR}(G; p)$ described in Example 3.6(b). For another graph $F$ on $m$ unordered vertices, Definition 4.3 introduces the metric $M_\infty(\mathrm{CR}(G; p), \mathrm{CR}(F; q))$ as the maximum of three distances. If $n > 1$, all signs in Definition 3.3 are zeros. Because $D(p) = -|p|$, the metric $M_\infty$ equals the maximum of $||p| - |q||$ and the bottleneck distance $W_\infty$ between the fixed clouds of unordered points $\{ (-|p'|, d(p', p)) \mid p' \in V(G) - \{p\} \}$ and $\{ (-|q'|, d(q', q)) \mid q' \in V(F) - \{q\} \}$ in $\mathbb{R}^2$. The weighted graph $\Gamma(G, F)$ has $m + m$ vertices associated with $p \in V(G)$ and $q \in V(F)$ with weights $M_\infty(\mathrm{CR}(G; p), \mathrm{CR}(F; q))$ on corresponding edges. By Definition 4.5 the final metric $\mathrm{NBM} = \mathrm{BMD}(\Gamma(G, F))$ equals the maximum weight $M_\infty(\mathrm{CR}(G; p), \mathrm{CR}(F; \beta(p)))$ minimized for a bijection $\beta : V(G) \to V(F)$.* ∎

The metrics $W_\infty, M_\infty, \mathrm{NBM}$ compare objects of the same size. To compare graphs with different numbers of vertices, $M_\infty$ in Definition 4.5 can be replaced with Earth Mover's Distance EMD Rubner et al. (2000), see Definition B.2, which satisfies all metric axioms. The axioms of all metrics and main Theorem 4.7 below are proved in appendices B and C.

**Theorem 4.7** (NCD solves Problem 1.1). *(a) The Nested Centered Distribution $\mathrm{NCD}(G; h)$ in Definition 3.5 is invariant under any rigid motion for all Euclidean graph $G$ on $m$ unordered vertices and, for a fixed dimension $n$, can be computed in time $O(n^2 m^{h+1})$ for any order $h \geq 1$.*

*(b) $\mathrm{NCD}(G; n)$ is a complete invariant of graphs $G \subset \mathbb{R}^n$ under rigid motion from the group $\mathrm{SE}(n)$.*

*(c) Perturbing each vertex of a graph $G \subset \mathbb{R}^n$ within its $\varepsilon$-neighborhood changes $\mathrm{NCD}(G; h)$ up to $2\varepsilon$ in both metrics NBM, EMD for any $h \geq 1$.*

*(d) For any graphs $G, F \subset \mathbb{R}^n$ on $m$ unordered vertices, the metrics NBM and EMD between the invariants $\mathrm{NCD}(G; h)$ and $\mathrm{NCD}(F; h)$ can be computed in time $O(m^{2h+1.5} \log^{h+1} m)$.* ∎

Theorem 4.7(b) means that any graphs $G, F \subset \mathbb{R}^n$ are related by rigid motion *if and only if* $\mathrm{NCD}(G; n) = \mathrm{NCD}(F; n)$. This equality is interpreted as a bijection $\mathrm{NCD}(G; n) \to \mathrm{NCD}(F; n)$

matching all CDs, which is equivalent to NBM $= 0$ by the first metric axiom. Since every CR can be stored in a vector form, the complete invariant $\text{NCD}(G; n)$ can be considered vectorial.

Table 2 emphasizes that most graphs should be first compared (or represented for machine learning) by simpler and faster invariants, so the complete $\text{NCD}(G; n)$ is used only in rare cases but is still necessary to make really important conclusions as we show later for all molecules in QM9.

Table 2: Hierarchy of invariants of $G \subset \mathbb{R}^2$ on $m$ unordered vertices: from the fastest to complete.

| invariant | $\text{SRV}(G)$ | $\text{SDV}(G)$ | $\text{PDD}(G)$ | $\text{NCD}(G; 1)$ | $\text{NCD}(G; 2)$ |
|---|---|---|---|---|---|
| time | $O(m \log m)$ | $O(m^2)$ | $O(m^2 \log m)$ | $O(m^2)$ | $O(m^3)$ |
| metric | $L_\infty$ | $L_\infty$ | EMD | NBM | NBM |
| time | $O(m)$ | $O(m^2)$ | $O(m^3)$ | $O(m^{3.5} \log^2 m)$ | $O(m^{5.5} \log^3 m)$ |

**Example 4.8** (Proof of Theorem 4.7(b) for $n = 1$). *For a graph $G \subset \mathbb{R}$ with the center of mass $O(G) = 0$, a base sequence $A$ from Definition 3.3 can be any vertex $p \in G$. Then $\text{sign}(p)$ is the usual sign of $p \in \mathbb{R}$, $D(p)$ is the signed distance $-|p|$, $R(G; p)$ is the $2 \times (m - 1)$ matrix whose column for any vertex $q \in V(G) - \{p\}$ consists of the distances $d(q, 0) = -|q|$ and $d(q, p) = \pm|q - p|$, where the plus sign $+$ indicates an edge between $q, p$, while the minus sign $-$ means no edge.*

*So $\text{CR}(G; p) = [\text{sign}(p), -|p|, R(G; p)]$. Every $R(G; p)$ uniquely determines $p$ in the line $\mathbb{R}$ by $\text{sign}(p)$ and $|p|$, and the position of any other vertex $q \in V(G) - \{p\}$ by its Euclidean distances $|q|$, $|q - p|$ to the origin and fixed vertex $p$. The position of any $q \in \mathbb{R}$ is uniquely determined by $\text{sign}(q)$ and $|q|$, which helps us unambiguously identify its Centered Representation $\text{CR}(G; q)$ in the unordered collection $\text{NCD}(G; 1)$ of all these CRs. The signs of $d(q, q')$ in each $R(G; q)$ determine the presence or absence of an edge of $G$ between any vertices $q, q' \in V(G)$.* ∎

## 5 EXPERIMENTS ON 130K+ MOLECULES, LIMITATIONS, AND SIGNIFICANCE

This section describes how the new invariants have enabled a complete classification of all molecular graphs in the QM9 database of 130K+ (130,808) molecules given with atomic 3D coordinates.

For graphs $G \subset \mathbb{R}^n$ with edges, such a practical classification was previously impossible because there was no complete and Lipschitz continuous invariant of Euclidean graphs even in $\mathbb{R}^2$. After the SSS theorem since 300 BC, such an invariant solving Problem 1.1 was known only for cyclic polygons inscribed into a circle of a fixed radius, see Theorem 1.8 on p.52 in Penner (2012). The complete invariant SCD of point clouds Widdowson & Kurlin (2023) has no nested structure of the new invariant NCD, which was needed to reconstruct all edges of $G \subset \mathbb{R}^n$ from $\text{NCD}(G; n)$. All experiments were done in 5 hours on Ryzen 9 3950X 3.5 GHz, 64 MB of L3 cache, RAM 82GB.

**The hierarchy of invariants** described in Fig. 3 and Table 2 justifies the computations starting from the simple and fast invariants on all pairs to filter out distant graphs and then moving to progressively stronger invariants for much smaller subsets of pairs of close molecules. We computed the pseudo-metric $L_\infty$ (the maximum absolute difference of corresponding coordinates) on all 873,527,974 pairs of SRVs, then 8,735,279 $L_\infty$ on the stronger SDVs for the 1% closest pairs, then 87,352 EMDs on PDDs for the 1% closest pairs, 10K distances NBM on $\text{NCD}(G; 1)$, $\text{NCD}(G; 2)$s for the top closest pairs, and 64 NBMs on complete invariants $\text{NCD}(G, 3)$ for molecular graphs $G \subset \mathbb{R}^3$.

Table 3: Closest chemically different molecules by distances in Å $= 10^{-10}$m, see Fig. 4 (right).

| invariant | metric | distance | molecule A | molecule B | composition A | composition B |
|---|---|---|---|---|---|---|
| $L_\infty$ | $\text{SRV}(G)$ | 0.02057 | 131923 | 5365 | H3 C4 N3 O2 | H4 C5 N2 O1 |
| $L_\infty$ | $\text{SDV}(G)$ | 0.05505 | 123533 | 24547 | H3 C4 N5 | H3 C5 N3 O1 |
| EMD | $\text{PDD}(G)$ | 0.05145 | 123533 | 24521 | H3 C4 N5 | H3 C5 N3 O1 |
| NBM | $\text{NCD}(G,3)$ | 0.07054 | 123532 | 24513 | H4 C5 N4 | H4 C6 N2 O1 |

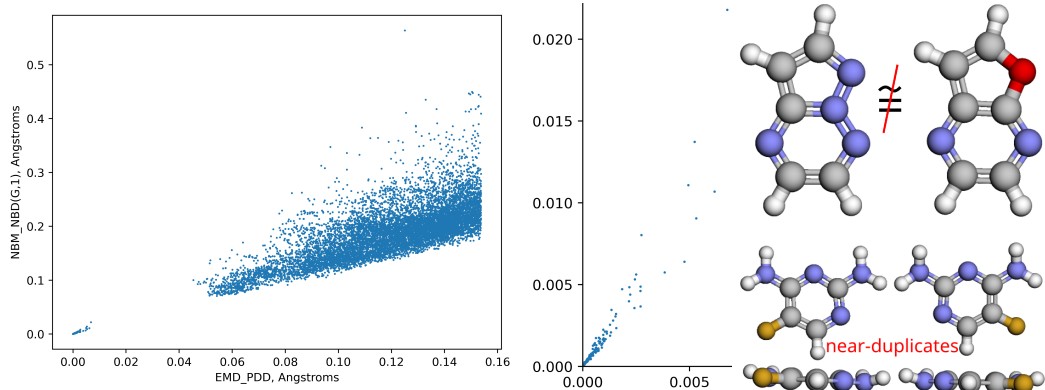

Figure 4: **Left**: each dot is a comparison of closest molecular graphs $G, F$ from QM9 by the pseudo-metric $x = \text{EMD}(\text{PDD}(G), \text{PDD}(G))$ vs $y = \text{NBM}(\text{NCD}(G; 1), \text{NCD}(F; 1))$. **Middle**: zoomed-in comparisons for small distances. **Top right**: the smallest $\text{NBM} \approx 0.07\text{Å}$ on $\text{NCD}(G; 3)$ for chemically different molecules is for 123533 and 24521. **Bottom right**: near-duplicate (almost flat) molecules 123532 and 24513 have the same composition and tiny $\text{EMD} \approx 2.37 \times 10^{-7}\text{Å}$ (not distinguishing mirror images) but $100\times$ higher $\text{NBM} \approx 2.95 \times 10^{-5}\text{Å}$ on complete $\text{NCD}(G; 3)$.

Fig. 5 shows that many pairs of molecular graphs have the distances NBM between their three invariants $\text{NCD}(G; h)$ for $h = 1, 2, 3$, which justifies faster computations for smaller $h$, but the complete invariant $\text{NCD}(G; 3)$ still better differentiates graphs than the invariants for $h = 1, 2$.

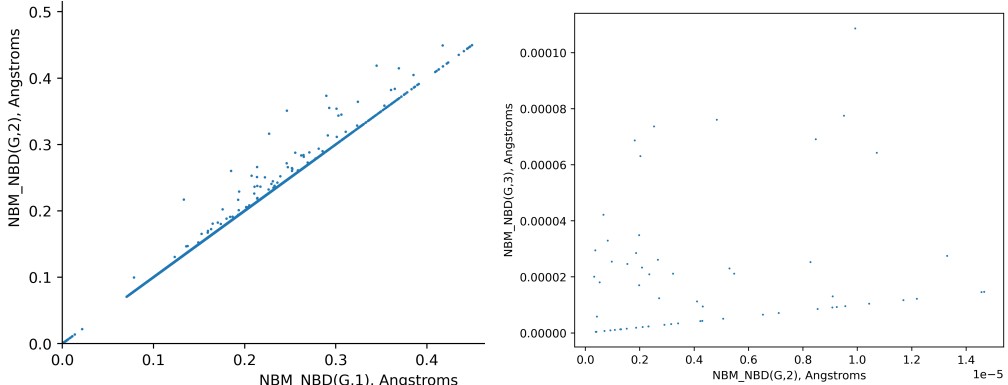

Figure 5: Each dot is a comparison of molecular graphs from QM9 by the distances on the progressively stronger invariants. **Left**: $\text{NCD}(G; 1)$ vs $\text{NCD}(G; 2)$. **Right**: $\text{NCD}(G; 2)$ vs $\text{NCD}(G; 3)$.

**Main conclusion** of all pairwise comparisons of molecular graphs from QM9 in a hierarchical way: all chemically different molecules are rigidly different, see the smallest distance $\text{NBM} \approx 0.07\text{Å}$ on complete invariants in Table 3. In other words, the map $\{\text{molecules}\} \rightarrow \{\text{graphs on atomic centers (without chemical elements)}\}$ is injective on rigid classes. Hence this map can be theoretically inverted on its image, so we can reconstruct all chemistry from sufficiently precise geometry.

The solution to Problem 1.1 settled the long-standing challenge of properly defining a *molecular structure*. A traditional approach including the recent one in Lang et al. (2023) is to describe such a structure as 'a set of unlabeled configurations that are relatively similar to each other'. This 'relative similarity' cannot depend on manual thresholds, which make any classification trivial as explained in Anosova et al. (2024), but should be rigorously defined as a class under equivalence (say, rigid motion) satisfying all axioms including the transitivity, which guarantees well-defined disjoint classes.

Even more importantly, all molecules of $m$ atoms have different and uniquely defined locations in a common continuous *Graph Rigid Space* $\text{GRS}(\mathbb{R}^3; m)$. The complete invariants $\text{NCD}(G; 3)$ provide exact geographic-style coordinates of any $G$ in this space as for any (say) houses on our

planet. The question of whether to put close neighbors like near-duplicates in Fig. 4(bottom right) into one cluster of the "same" molecules is rather administrative (similar to putting close houses into one village instead of different ones) for domain experts rather than purely scientific.

Studying molecules by fixing a composition is similar to drawing artificial boundaries between countries on Earth. Because some molecules of different compositions have close shapes as in Fig. 4(top right), they should have similar properties. Now any properties of molecules should be possible to predict only from complete invariants $\mathrm{NCD}(G; 3)$ even without chemistry in the same way as any precise geographic location uniquely determines all physical properties of this place such as the average annual temperature. Chemical compositions can be still helpful similar to the location's altitude, which easier predicts this temperature than theoretically sufficient geographic coordinates.

Fig. 6 (left) shows the simplest geographic-style map of QM9 as a finite sample within $\bigcup_{m=3}^{29} \mathrm{GRS}(\mathbb{R}^3; m)$ projected to the invariants $\mathrm{SRV}_1 \geq \mathrm{SRV}_2$. All molecules on the horizontal axis $y = \mathrm{SRV}_1 - \mathrm{SRV}_2 = 0$ have $\mathrm{SRV}_1 = \mathrm{SRV}_2$ (due to two equidistant atoms from the center of mass) and can be projected (like any subset of QM9) to other coordinates as in Fig. 6 (right). Molecular properties can be visualized on these geographic maps as 'mountainous' landscapes.

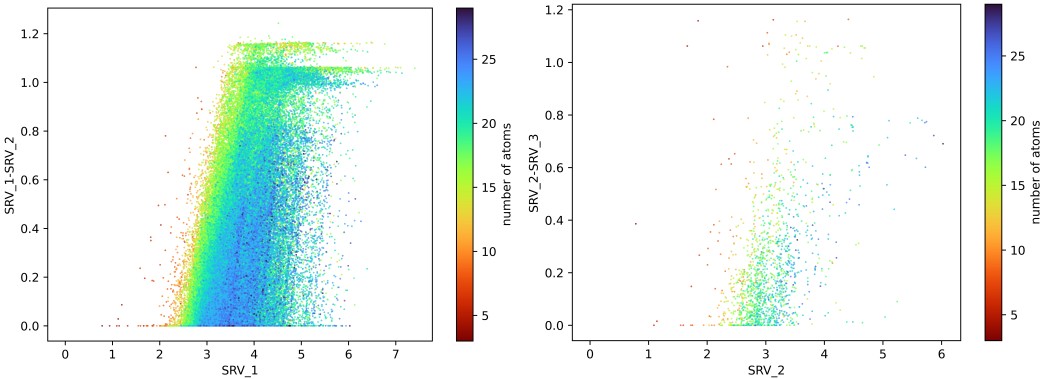

Figure 6: **Left**: every dot represents a molecular graph with the invariant coordinates $x = \mathrm{SRV}_1$, $y = \mathrm{SRV}_1 - \mathrm{SRV}_2$, all in Angstroms, where $1\text{Å} = 10^{-10}m \approx$ the smallest interatomic distance. **Right**: projection of QM9 to $x = \mathrm{SRV}_1$, $y = \mathrm{SRV}_2 - \mathrm{SRV}_3$. The color is by the number of atoms.

The limitation is the time $O(m^{2h+1.5} \log^{h+1} m)$ of the metric NBM, which is better than exponential and practical for $h = 1, 2, 3$ up to 29 atoms in QM9. The hierarchy of $\mathrm{NCD}(G; h)$ allows us to filter out distant graphs by faster distances starting with $L_\infty$ on $\mathrm{NCD}(G; 0) = \mathrm{SRV}(G)$ in $O(m)$ time. Theorem 4.7 is harder than (Widdowson & Kurlin, 2023, Theorem 4.7) for point clouds because exponentially many different graphs can have the same vertex set, see Fig. 1.

Any vertex $p$ and edge of $G$ can have an *attribute* and a *weight* respected by any isometry that maps one graph to another. These vertex attributes and edge weights can be incorporated as extra columns and rows in CRs from Definition 3.3, and then incorporated into NCD and NBM. We can compare graphs of different numbers of vertices because EMD works for both PDD and NCD as weighted distributions of any finite size. This comparison splits the vertices from $V(G)$ into parts (subvertices) that are optimally 'transported' to a splitting of another vertex set $V(F)$.

Already for $m = 4$ unordered points in $\mathbb{R}^2$, there was no parametrized map for a 5-dimensional space of (isometry classes of) plane quadrilaterals. Indeed, 6 pairwise distances between 4 ordered points in $\mathbb{R}^2$ uniquely determine such a 4-point cloud uniquely up to isometry due to (Dekster & Wilker, 1987, Theorem 1) but these 6 distances should satisfy a complicated polynomial equation saying that the tetrahedron on 4 points has volume 0. Hence random inter-point distances are realizable by a cloud of $m > n + 1$ unordered points in $\mathbb{R}^n$ with probability 0 Duxbury et al. (2016).

Problem 1.1 provides a roadmap for the discriminative (and then generative) approach to any data objects by replacing graphs and isometries with other data and equivalences. The supplementary materials include the code and all proofs. We thank all reviewers for their valuable time and advice.

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

## A    EXTRA EXPERIMENTS ON MOLECULAR GRAPHS

Past maps of QM9 in Fig. 7 based on eigenvalues are too dense without clear separation. Even if we zoom in, these two or three incomplete invariants will not provide any extra separation. The complete invariants NDP contain much more geometric information.

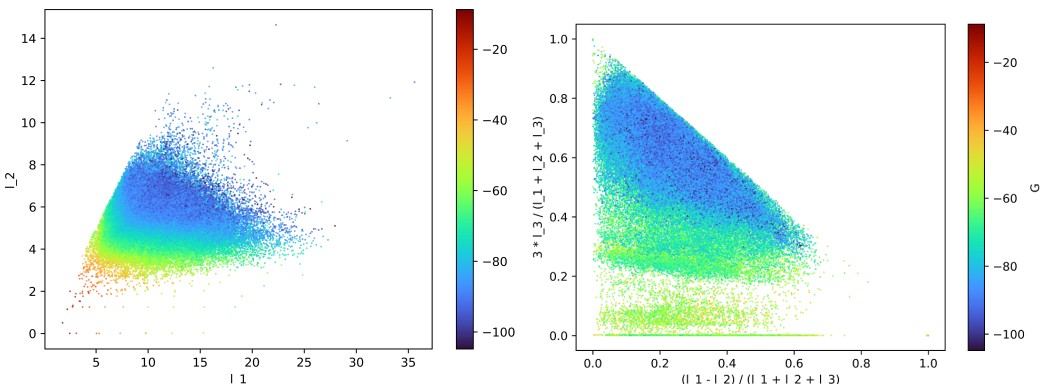

Figure 7: **Left**: each dot represents one QM9 molecule whose atomic cloud has two largest roots $l_1 \geq l_2$ of eigenvalues (moments of inertia Nemec (2022) or elongations in two principal directions) in Angstroms ($1\mathring{A} = 10^{-10}m \approx$ smallest interatomic distance). The color represents the free energy $G$ characterizing molecular stability. **Right**: each dot represents one QM9 molecule whose atomic cloud has coordinates $x, y$ expressed via the roots $l_1 \geq l_2 \geq l_3 \geq 0$ of three eigenvalues.

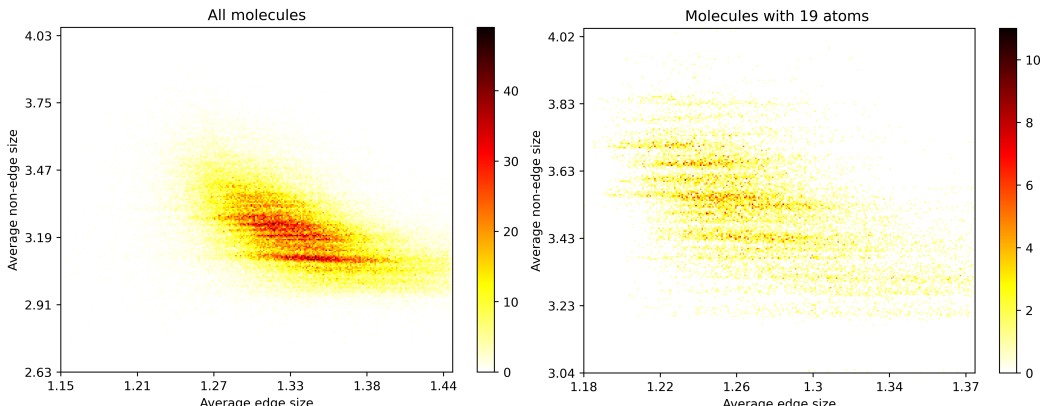

Figure 8: **Left**: the heatmap of all molecular graphs from QM9 in the simplest continuous invariants. **Right**: 18336 graphs with 19 atoms. The color indicates the number of molecules at every pixel.

## B    METRICS ON GRAPHS AND THEIR CONTINUITY UNDER PERTURBATIONS

This appendix verifies the axioms and Lipschitz continuity for auxiliary metrics in section 4.

**Lemma B.1** (metric axioms for the bottleneck matching distance BMD). *Let $S, Q$ be any unordered distributions of the same number of objects with a base metric $d$. Define the complete bipartite graph $\Gamma(S, Q)$ whose every edge $e$ joining objects $R_S \in S$ and $R_Q \in Q$ has the weight $w(e) = d(R_S, R_Q)$. Then the bottleneck matching distance $\mathrm{BMD}(\Gamma(S, Q))$ from Definition 4.4 satisfies all metric axioms on such unordered distributions.*

***Proof of Lemma** B.1.* The coincidence axiom means that $\mathrm{NBM}(S, Q) = 0$ if and only if the weighted distributions $S, Q$ are equal in the sense that there is a bijection $g : S \rightarrow Q$ so that $d(g(R), R) = 0$ for any $R \in S$.

Indeed, if the weighted distributions $S, Q$ can be matched by a bijection, we get a vertex matching $E$ of $\Gamma(S, Q)$ whose all edges have weights $w(e) = 0$. Definition 4.4 implies that $\text{BMD}(\Gamma(S, Q)) = 0$ as required.

Conversely, if $\text{BMD}(\Gamma(S, Q)) = 0$, there is a vertex matching $E$ in $\Gamma(S, Q)$ with all $w(e) = 0$. This matching $E$ defines a required bijection $S \to Q$. The symmetry $\text{BMD}(\Gamma(S, Q)) = \text{BMD}(\Gamma(Q, S))$ follows from Definition 4.4 and the symmetry of the base metric $d$.

To prove the triangle inequality

$$\text{BMD}(\Gamma(S, Q)) + \text{BMD}(\Gamma(Q, T)) \geq \text{BMD}(\Gamma(S, T)),$$

let $E_{SQ}, E_{QT}$ be optimal vertex matchings in the graphs $\Gamma(S, Q), \Gamma(Q, T)$, respectively, such that

$$\text{BMD}(\Gamma(S, Q)) = W(E_{SQ}), \text{BMD}(\Gamma(Q, T)) = W(E_{QT}),$$

see Definition 4.4. The composition $E_{SQ} \circ E_{QT}$ is a vertex matching in $\Gamma(S, T)$, so $W(E_{SQ} \circ E_{QT}) \geq \text{BMD}(\Gamma(S, T))$. It suffices to prove that

$$W(E_{SQ}) + W(E_{QT}) \geq W(E_{SQ} \circ E_{QT}).$$

Let $e_{ST}$ be an edge with a largest weight from $E_{SQ} \circ E_{QT}$, so $W(E_{SQ} \circ E_{QT}) = w(e_{ST})$. The edge $e_{ST}$ can be considered the union of edges $e_{SQ} \in E_{SQ}, e_{QT} \in E_{QT}$.

By the triangle inequality for the base metric $d$,

$$w(e_{SQ}) + w(e_{QT}) \geq w(e_{ST}) = W(E_{SQ} \circ E_{QT})$$

implies that

$$W(E_{SQ}) + W(E_{QT}) \geq W(E_{SQ} \circ E_{QT})$$

because both terms on the left-hand side are maximized for all edges (not only $e_{SQ}, e_{QT}$) from $E_{SQ}, E_{QT}$. $\qquad\square$

Definition B.2 below makes sense for any distributions $\{[R_1, w_1], \ldots, [R_m, w_m]\}$, where $R_1, \ldots, R_m$ are objects with a base metric $d$ and weights $w_1, \ldots, w_m \in [0, 1]$. Each $R_i$ can be CBR or CBD of any depth with a base metric $M_\infty$ or BMD from Definitions 4.3, 4.5.

**Definition B.2** (EMD). *Let $S = \{[R_i(S), w_i(S)]\}_{i=1}^{m(S)}$ and $Q = \{[R_j(Q), w_j(Q)]\}_{j=1}^{m(Q)}$ be weighted distributions of objects $R_i(S), R_j(Q)$, which live in a space with a metric $d$. A* flow *from $S$ to $Q$ is an $m(S) \times m(Q)$ matrix whose element $f_{ij} \in [0, 1]$ represents a* partial flow *from $R_i(S)$ to $R_j(Q)$. The* Earth Mover's Distance *is the minimum cost $\text{EMD}(S, Q) = \sum_{i=1}^{m(S)} \sum_{j=1}^{m(Q)} f_{ij} d(R_i(S), R_j(Q))$ for variable 'flows' $f_{ij} \in [0, 1]$ subject to the conditions $\sum_{j=1}^{m(Q)} f_{ij} \leq w_i(S)$ for $i = 1, \ldots, m(S)$, $\sum_{i=1}^{m(S)} f_{ij} \leq w_j(Q)$ for $j = 1, \ldots, m(Q)$, and $\sum_{i=1}^{m(S)} \sum_{j=1}^{m(Q)} f_{ij} = 1$.*

The first condition $\sum_{j=1}^{m(Q)} f_{ij} \leq w_i(S)$ means that not more than the weight $w_i(S)$ of $R_i(S)$ 'flows' into all $R_j(Q)$ via 'flows' $f_{ij}$, $j = 1, \ldots, m(Q)$. The second condition $\sum_{i=1}^{m(S)} f_{ij} = w_j(Q)$ means that all 'flows' $f_{ij}$ from $R_i(S)$ for $i = 1, \ldots, m(S)$ 'flow' into $R_j(Q)$ up to the maximum weight $w_j(Q)$. The last condition $\sum_{i=1}^{m(S)} \sum_{j=1}^{m(Q)} f_{ij} = 1$ forces to 'flow' all rows $R_i(S)$ to all rows $R_j(Q)$.

The EMD satisfies all metric axioms, see the appendix in Rubner et al. (2000), needs $O(m^3 \log m)$ time for distributions of a maximum size $m$ and is approximated in $O(m)$ time, see Shirdhonkar & Jacobs (2008); Sato et al. (2020).

Definition B.2 can be adapted for the EMD between NDDs by (1) replacing the bottleneck distance $W_\infty$ in Definition 4.3 with EMD between clouds of equally weighted points, and (2) replacing

BMD($\Gamma$) for a bipartite graph $\Gamma$ with EMD($\Gamma$) between the unordered sets (of potentially different sizes) of BDDs with weights on all white vertices and BDDs on all black vertices.

The Lipschitz continuity of NDD and EMD in Theorem 4.7(c) needs Lemmas B.3, B.4, C.8.

If a metric graph $G$ lives in an ambient metric space $X$, a natural perturbation of $G$ is a shift of every vertex of $G$ up to $\varepsilon$ in the metric of $X$. Then the distance $d(p, q)$ between any vertices $p, q$ of $G$ changes by at most $2\varepsilon$.

We will prove the continuity in more general settings by only assuming that $d(p, q)$ changes by at most $2\varepsilon$ for any $p, q \in V(G)$ without requiring an ambient space $X$.

**Lemma B.3** (Lipschitz continuity of BMD). *Let $\Gamma$ be a complete bipartite graph with a vertex matching $E$ such that any $e \in E$ has a weight $w(e) \leq \varepsilon$. Then $\mathrm{BMD}(\Gamma) \leq \varepsilon$.*

***Proof of Lemma B.3.*** By Definition 4.4, the given matching $E$ has the weight $W(E) = \max_{e \in E} w(e) \leq \varepsilon$. Since $\mathrm{BMD}(\Gamma) = \min_E W(E)$ is minimized for all vertex matchings, we get $\mathrm{BMD}(\Gamma) \leq \varepsilon$. $\qquad\square$

**Lemma B.4** (Lipschitz continuity of EMD). *In Definition B.2, let distributions $S, Q$ have a bijection $R_i(S) \leftrightarrow R_i(Q)$ between equally weighted objects such that $d(R_i(S), R_i(Q)) \leq \varepsilon$ for all $i = 1, \ldots, m$, where $m = m(S) = m(Q)$. Then $\mathrm{EMD}(S, Q) \leq \varepsilon$.*

***Proof of Lemma B.4.*** In Definition B.2, choose partial flows $f_{ij} = \dfrac{1}{m}$ for $i = j$, otherwise $f_{ij} = 0$.

Then $\mathrm{EMD}(S, Q) \leq \sum\limits_{i=1}^{m} \sum\limits_{j=1}^{m} f_{ij} d(R_i(S), R_j(Q)) = \sum\limits_{i=1}^{m} \dfrac{1}{m} d(R_i(S), R_i(Q)) \leq \dfrac{1}{m} \sum\limits_{i=1}^{m} \varepsilon = \varepsilon.$ $\quad\square$

## C    PROOFS FOR EUCLIDEAN GRAPHS FROM SECTION 3

This appendix rigorously proves all parts of Theorem 4.7.

The *affine dimension* $0 \leq \mathrm{aff}(A) \leq n$ of a cloud $A = \{p_1, \ldots, p_m\} \subset \mathbb{R}^n$ is the maximum dimension of the vector space generated by all inter-point vectors $p_i - p_j$, $i, j \in \{1, \ldots, m\}$. Then $\mathrm{aff}(A)$ is an isometry invariant and is independent of an order of points of $A$. Any cloud $A$ of 2 distinct points has $\mathrm{aff}(A) = 1$. Any cloud $A$ of 3 points that are not in the same straight line has $\mathrm{aff}(A) = 2$.

Lemma C.1 provides a simple criterion for a matrix to be realizable by squared distances of a point cloud in $\mathbb{R}^n$.

**Lemma C.1** (realization of distances). *(a) A symmetric $m \times m$ matrix of $s_{ij} \geq 0$ with $s_{ii} = 0$ is realizable as a matrix of squared distances between points $p_0 = 0, p_1, \ldots, p_{m-1} \in \mathbb{R}^n$ if and only if the $(m-1) \times (m-1)$ matrix $g_{ij} = \dfrac{s_{0i} + s_{0j} - s_{ij}}{2}$ has only non-negative eigenvalues.*

*(b) If the condition in (a) holds, $\mathrm{aff}(0, p_1, \ldots, p_{m-1})$ equals the number $k \leq m - 1 \leq n$ of positive eigenvalues. Also in this case, $g_{ij} = p_i \cdot p_j$ define the* Gram matrix GM *of the vectors $p_1, \ldots, p_{m-1} \in \mathbb{R}^n$, which are uniquely determined in time $O(m^3)$ up to an orthogonal map in $\mathbb{R}^n$.*

***Proof of Lemma C.1.*** **(a)** We extend Theorem 1 from Dekster & Wilker (1987) to the case $m < n + 1$ and justify the reconstruction of $p_1, \ldots, p_{m-1}$ in time $O(m^3)$ uniquely in $\mathbb{R}^n$ up to an orthogonal map from $\mathrm{O}(n)$.

The part *only if* $\Rightarrow$. Let a symmetric matrix $S$ consist of squared distances between points $p_0 = 0, p_1, \ldots, p_{m-1} \in \mathbb{R}^n$. For $i, j = 1, \ldots, m - 1$, the matrix with the elements

$$g_{ij} = \frac{s_{0i} + s_{0j} - s_{ij}}{2} = \frac{p_i^2 + p_j^2 - |p_i - p_j|^2}{2} = p_i \cdot p_j$$

is the Gram matrix, which can be written as $\mathrm{GM} = P^T P$, where the columns of the $n \times (m-1)$ matrix $P$ are the vectors $p_1, \ldots, p_{m-1}$. For any vector $v \in \mathbb{R}^{m-1}$, we have

$$0 \leq |Pv|^2 = (Pv)^T(Pv) = v^T(P^T P)v = v^T \mathrm{GM} v.$$

Since the quadratic form $v^T \mathrm{GM} v \geq 0$ for any $v \in \mathbb{R}^{m-1}$, the matrix GM is positive semi-definite meaning that GM has only non-negative eigenvalues, see Theorem 7.2.7 in Horn & Johnson (2012).

The part *if* $\Leftarrow$. For any positive semi-definite matrix GM, there is an orthogonal matrix $Q$ such that $Q^T \mathrm{GM} Q = D$ is the diagonal matrix, whose $m-1$ diagonal elements are non-negative eigenvalues of GM. The diagonal matrix $\sqrt{D}$ consists of the square roots of eigenvalues of GM.

**(b)** The number of positive eigenvalues of GM equals the dimension $k = \mathrm{aff}(\{0, p_1, \ldots, p_{m-1}\})$ of the subspace in $\mathbb{R}^n$ linearly spanned by $p_1, \ldots, p_{m-1}$. We may assume that all $k \leq n$ positive eigenvalues of GM correspond to the first $k$ coordinates of $\mathbb{R}^n$. Since $Q^T = Q^{-1}$, the given matrix $\mathrm{GM} = QDQ^T = (Q\sqrt{D})(Q\sqrt{D})^T$ becomes the Gram matrix of the columns of $Q\sqrt{D}$. These columns become the reconstructed vectors $p_1, \ldots, p_{m-1} \in \mathbb{R}^n$.

If there is another diagonalization $\tilde{Q}^T \mathrm{GM} \tilde{Q} = \tilde{D}$ for $\tilde{Q} \in \mathrm{O}(n)$, then $\tilde{D}$ differs from $D$ by a permutation of eigenvalues, which is realized by an orthogonal map, so we set $\tilde{D} = D$. Then $\mathrm{GM} = \tilde{Q}D\tilde{Q}^T = (\tilde{Q}\sqrt{D})(\tilde{Q}\sqrt{D})^T$ is the Gram matrix of the columns of $\tilde{Q}\sqrt{D}$.

The new columns differ from the previously reconstructed vectors $p_1, \ldots, p_{m-1} \in \mathbb{R}^n$ by the orthogonal map $Q\tilde{Q}^T$. Hence the reconstruction is unique up to $\mathrm{O}(n)$-transformations. Computing eigenvectors $p_1, \ldots, p_{m-1}$ requires a diagonalization of GM in time $O(m^3)$ (Press et al., 2007, section 11.5). $\qquad\square$

Though Lemma C.1 gives a two-sided criterion for realizability of distances by points $p_1, \ldots, p_m \in \mathbb{R}^n$, the space of distance matrices is highly singular and cannot be easily sampled. Even $m = 4$ points in $\mathbb{R}^2$ have 6 distances that should satisfy a polynomial equation saying that the tetrahedron with these 6 edge lengths has volume 0. So a randomly sampled matrix of potential distances for $m > n+1$ is unlikely to be realizable by a cloud of $m$ ordered points in $\mathbb{R}^n$.

Chapter 3 in Liberti & Lavor (2017) discusses realizations of a complete graph given by a distance matrix in $\mathbb{R}^n$. Lemma C.2(a) and later results hold for all clouds including degenerate ones, e.g. for 3 points in a straight line.

Any points $p_1, \ldots, p_{n-1} \in A$ have $\mathrm{aff}(p_1, \ldots, p_{n-1}) \leq n-2$. For example, any two distinct points in $A \subset \mathbb{R}^3$ generate a straight line. In $\mathbb{R}^2$, any point $p_1 \neq O(A)$ forms a suitable $\{p_1\}$. In $\mathbb{R}^3$, one can choose any distinct points $p_1, p_2 \in A$ so that the infinite straight line via $p_1, p_2$ avoids $O(A)$.

If there are no such $p_1, p_2$, then $A \subset \mathbb{R}^3$ is contained in a straight line $L$, so $\mathrm{aff}(A) = 1$. In this degenerate case, the stronger condition $\mathrm{aff}(O(A) \cup \{p_1, \ldots, p_{n-1}\}) = \mathrm{aff}(A)$ will help reconstruct $A \subset L$ by using any point $p_1 \neq O(A)$. The first step is to reconstruct any ordered sequence from its distance matrix in Lemma C.2(a).

Lemma C.2(a) holds for all degenerate clouds, e.g. for three points are in a straight line.

**Lemma C.2** (reconstruction of ordered points). *(a) Any sequence of ordered points $A = (p_1, \ldots, p_m)$ in $\mathbb{R}^n$ can be reconstructed (uniquely up to isometry) from the matrix of the Euclidean distances $|p_i - p_j|$ in time $O(m^3)$. If all distances are divided by $R = \max_{i=1,\ldots,m} |p_i|$, the reconstruction of $A \subset \mathbb{R}^n$ is unique up to isometry and uniform scaling.*

*(b) If $m \leq n$, the uniqueness of reconstructions in part (a) holds if we replace isometry with rigid motion. Hence any $n-1$ ordered points $p_1, \ldots, p_{n-1}$ can be uniquely reconstructed from all pairwise distances between $0, p_1, \ldots, p_{n-1}$ up to $\mathrm{SO}(n)$ rotation around the origin $0 \in \mathbb{R}^n$.*

***Proof of Lemma*** *C.2.* **(a)** By translation, we can put $p_1$ at the origin $0 \in \mathbb{R}^n$. Let GM be the $(m-1) \times (m-1)$ matrix $g_{ij} = \dfrac{p_i^2 + p_j^2 - |p_i - p_j|^2}{2} = p_i \cdot p_j$ constructed from squared distances between

$p_1 = 0, \ldots, p_m$ for $i, j = 2, \ldots, m$. By Lemma C.1(b) if GM has $k \leq n$ positive eigenvalues, then $p_1 = 0, \ldots, p_m$ can be uniquely determined up to isometry in $\mathbb{R}^k \subset \mathbb{R}^n$ in time $O(m^3)$. If all distances are divided by the same radius $R$, the above construction guarantees uniqueness up to isometry and uniform scaling.

**(b)** If $m \leq n$, any mirror image of $A \subset \mathbb{R}^n$ after a suitable rigid motion in $\mathbb{R}^n$ can be assumed to belong to an $(n-1)$-dimensional hyperspace $H \subset \mathbb{R}^n$, where they are matched by a mirror reflection $H \to H$ with respect to an $(n-2)$-dimensional subspace $S \subset H$. This reflection is realized by the $\mathrm{SO}(n)$ rotation through $180°$ around $S$. $\qquad\square$

Lemma C.2(b) for $m = n = 3$ implies that any triangle is determined by its sides up to rigid motion in $\mathbb{R}^3$. For example, the sides $3, 4, 5$ define a right-angled triangle whose mirror images are not related by rigid motion inside a plane $H \subset \mathbb{R}^3$, but are matched by composing a suitable rigid motion in $H$ and a $180°$ rotation of $\mathbb{R}^3$ around a line in $H$.

**Lemma C.3** (time of determinant). *Any $n \times n$ determinant can be computed in time $O(n^3)$.*

***Proof of Lemma C.3.*** Any $n \times n$ determinant can be computed by Gaussian elimination in time $O(n^3)$, see Bunch & Hopcroft (1974). The more recent theoretical estimate is $O(n^{2.373})$ by Fisikopoulos & Penaranda (2016). $\qquad\square$

***Proof of Theorem 4.7(a).*** Any rigid motion of $\mathbb{R}^n$ mapping a Euclidean graph $G \subset \mathbb{R}^n$ to another graph $F$ is a bijection preserving distances and signs of determinants, and hence induces a bijection $\mathrm{CBR}(G; p_1, \ldots, p_i) \to \mathrm{CBR}(F; q_1, \ldots, q_i)$ for all $p_1, \ldots, p_i \in V(G)$ and corresponding vertices $q_1, \ldots, q_i \in V(F)$ for any $i = 1, \ldots, h$, which implies a bijection $\mathrm{NCD}(G; h) \to \mathrm{NCD}(F; h)$. By Definition 3.5, if $G$ has $m$ unordered vertices, the $\mathrm{NCD}(G)$ consists of $m(m-1) \ldots (m-h+1) = O(m^h)$ Centered Base Representations $\mathrm{CBR}(G; A)$ for all base sequences $A \subset V(G)$ of $h$ ordered vertices.

Every $\mathrm{CBR}(G; A)$ consists of the three components $\mathrm{sign}(A), \mathrm{CD}(A), \mathrm{CR}(G; A)$. For $h = n$, $\mathrm{sign}(A)$ is the $n \times n$ determinant computable in time $O(n^3)$ by Lemma C.3. The distance matrix $\mathrm{CD}(A)$ needs $O(h^2)$ time. The $(h+1) \times (m-h)$ matrix $\mathrm{CR}(G; A)$ has $O(hm)$ distances, each computable in time $O(n)$. So $\mathrm{CBR}(G; A)$ can be computed in time $O(n^2 m)$ for $n \leq m$. Multiplying this time by the number $O(m^h)$ of base sequences gives the final time $O(n^2 m^{h+1})$ for $\mathrm{NCD}(G)$. $\qquad\square$

The proof of Theorem 4.7(b) will use the fact that any point in $\mathbb{R}^n$ is uniquely determined by $n+1$ distances to $n+1$ ordered points that affinely span $\mathbb{R}^n$, and also Lemma C.4.

**Lemma C.4** (equal CBRs). *Let a Euclidean graph $G \subset \mathbb{R}^n$ have the vertex set $V(G)$ with the center of mass at $p_0 = 0 \in \mathbb{R}^n$. Let $n-1$ ordered vertices $p_1, \ldots, p_{n-1}$ linearly span an $(n-1)$-dimensional subspace $S \subset \mathbb{R}^n$. Let $G(p_1, \ldots, p_{n-1})$ be the subgraph of $G$ on the vertex set $V(G)$ and all edges of $G$ at $p_1, \ldots, p_{n-1}$. For any other vertex $p$, let $\mathrm{CBR}'(G; p_1, \ldots, p_{n-1}, p)$ be obtained from the Centered Base Representation $\mathrm{BR}(G; p_1, \ldots, p_{n-1}, p)$ by removing signs of distances from all vertices $q \in V(G) - \{p_1, \ldots, p_{n-1}, p\}$ to $p$. If $\mathrm{BR}'(G; p_1, \ldots, p_{n-1}, p) = \mathrm{BR}'(G; p_1, \ldots, p_{n-1}, p')$ for some vertices $p, p' \in V(G) - \{p_1, \ldots, p_{n-1}\}$, the mirror reflection with respect to $S$ maps $G(p_1, \ldots, p_{n-1})$ to itself and $p$ to $p'$.*

***Proof of Lemma C.4.*** Under the reflection $f_S$ of $\mathbb{R}^n$ with respect to the subspace $S \subset \mathbb{R}^n$, the vertices $p, p'$ should be swapped because they have equal (signed) distances to the ordered points $p_0, \ldots, p_{n-1} \in S$. The equality of given $\mathrm{CBR}'$s means that $V' = V(G) - \{p_1, \ldots, p_{n-1}, p, p'\}$ bijectively maps to itself via $q \mapsto q'$ so that any matched $q, q'$ have the same distances to the $n+1$ ordered points $p_0, \ldots, p_{n-1}, p$ as to $p_0, \ldots, p_{n-1}, p'$, respectively. Any point in $\mathbb{R}^n$ is determined by its distances to the $n$ affinely independent points $p_0, \ldots, p_{n-1}$ up to the mirror reflection $f_S$. Since $f_S$ fixes $p_0, \ldots, p_{n-1}$, the reflection $f_S$ should swap $q, q'$ in such pairs and all their edges, so we conclude that $f_S(G(p_1, \ldots, p_{n-1})) = G(p_1, \ldots, p_{n-1})$ and $f_S(p) = p'$. $\qquad\square$

***Proof of Theorem 4.7(b).*** The completeness is proved by reconstructing any Euclidean graph $G \subset \mathbb{R}^n$ from $\mathrm{NCD}(G; n)$ uniquely up to rigid motion.

We prove that any Euclidean graph $G \subset \mathbb{R}^n$ can be reconstructed from its Nested Distance Distribution $\mathrm{NCD}(G; n)$ by induction on the dimension $n$.

The inductive base $n = 1$ is Example 4.8. Assume that any graph $G$ on $m$ unordered vertices can be reconstructed in $\mathbb{R}^k$ in time $O(k^3 m)$ for any $k < n$. Below we prove the inductive step for the dimension $n > 1$. Start from any $\mathrm{CBR}(G; A) = [\mathrm{sign}(A), \mathrm{CD}(A), \mathrm{CR}(G; A)]$ from Definition 3.5, where $A$ is a sequence of some $n$ ordered (not yet geometrically fixed) vertices $p_0, \ldots, p_n \in V(G)$. The first point $p_0$ is fixed at the origin $0 \in \mathbb{R}^n$ as usual by translation.

Lemma C.1(b) for the matrix $\mathrm{CD}(A)$ gives the number $k \le n$ of positive eigenvalues of the Gram matrix of the $n$ vectors $p_1, \ldots, p_n$ in time $O(n^3)$. If $\mathrm{aff}(A) = k < n$, we use the nested structure of $\mathrm{NCD}(G; n)$ to take another $\mathrm{CBR}(G; p_1, \ldots, p_k, q, \ldots, p_n)$ for a new vertex $q \in V(G) - A$. Check if $\mathrm{aff}(p_1, \ldots, p_k, q) = k + 1$ again by Lemma C.1(b) using the matrix $D(p_1, \ldots, p_k, q)$. If the affine dimension has not increased, we take another CBR with the same points $p_1, \ldots, p_k$ and a new $(k+1)$-st point from $V(G) - \{A \cup q\}$ and so on.

This search through Centered Base Representations involving the remaining vertices of $G$ requires a maximum of $m - n - 1$ steps with $O(n^3)$ time for every computation of the affine dimension. Hence in time $O(n^3 m)$, we can find a Centered Base Representation $\mathrm{CBR}(G; A)$ whose base sequence $A$ affinely generates the subspace of dimension $k = \mathrm{aff}(V(G))$ in $\mathbb{R}^n$. If $k < n$, the proof follows from the inductive hypothesis for the smaller dimension $k$.

If $\mathrm{aff}(V(G)) = n$, use the same notations for the fixed vertices $0 = p_0, \ldots, p_n$ that linearly generate $\mathbb{R}^n$. Lemma C.2(a) for $m = n + 1$ and the distance matrix $D(A)$ allow us to reconstruct $n + 1$ ordered points $0 = p_0, \ldots, p_n$ up to isometry in $\mathbb{R}^n$ in time $O(n^3)$. By Definition 3.5 every column of $\mathrm{CR}(G; A)$ contains Euclidean distances from the vertices $0 = p_0, \ldots, p_n \in \mathbb{R}^n$, which affinely generate $\mathbb{R}^n$, to another vertex $q \in V(G) - A$.

These $n + 1$ distances uniquely determine the position of $q$ in $\mathbb{R}^n$ whose coordinates can be found as follows. Each scalar product $q \cdot p_i$ can be computed as $|q| \cdot |p_i| \cos \angle(q, 0, p_i) = \dfrac{|q|^2 + |p_i|^2 - |q - p_i|^2}{2}$ for $i = 1, \ldots, n$. On another hand, $q \cdot p_i$ is a linear combination of unknown coordinates of $q$ with coefficients equal to the coordinates of $p_i$. One can find all coordinates of $q$ in time $O(n^3)$ by solving the system of linear equations, where the $n \times n$ determinant on the linear basis $p_1, \ldots, p_n$ is not zero. The total time is $O(n^3 m)$.

Since all vertices $q \in V(G) - A$ are geometrically unique, they can be (arbitrarily) ordered, say $p_{n+1}, \ldots, p_m$, following $p_0, \ldots, p_n$. The signs of distances in the matrix $\mathrm{CR}(G; A)$ also tell us about (present or absent) edges from $p_0, \ldots, p_n$ to all other vertices $q \in V(G) - A$.

The nested structure of $\mathrm{NCD}(G; n)$ allows us to consider $m - n$ unordered Base Representations $\mathrm{CBR}(G; p_1, \ldots, p_{n-1}, p_j)$ for all vertices $p_j$ with $j = n, \ldots, m$. Every vertex $p_j \in V(G)$ is uniquely determined in $\mathbb{R}^n$ by the column of its signed distances to $p_0, \ldots, p_n$ in the $(n+1) \times (m - n - 1)$ matrix $R(G; p_0, \ldots, p_n)$ for $j = n + 1, \ldots, m$.

By Lemma C.4, this distance list of $p_j$ (without edges between $p_j, p_k$ for $j, k > n$) suffices to identify one or maximum two Base Representations among all $m - n$ unordered CBRs with the fixed $n$ points $p_0, \ldots, p_{n-1}$ and variable $n$-th vertices. If there is a choice of two CBRs, we can take any of them for $p_j$. Indeed, choosing another vertex $p_k$, which should be mirror symmetric to $p_j$, will produce a mirror image of the reconstructed subgraph $G(p_1, \ldots, p_n, p_j)$ by Lemma C.4.

The matrix $\mathrm{CR}(G; p_1, \ldots, p_{n-1}, p_j)$ from the found CBRs contains signs that determine the (present or absent) edges from $p_j$ to all other vertices $p_k$ for $k = n + 1, \ldots, m$.

To guarantee the uniqueness of $G \subset \mathbb{R}^n$ under rigid motion and not only under isometry, we additionally use $\mathrm{sign}(p_1, \ldots, p_n)$ from CBR to fix an orientation of the simplex on $p_0, \ldots, p_n$. $\qquad\square$

The strength $\sigma(A)$ depends only on the distance matrix $D(A)$, we write $\sigma(A)$ for brevity. When the simplex on $A$ degenerates, the strength $\sigma(A)$ vanishes and is Lipschitz continuous by Lemma 4.2, while the volume of the simplex on $B$ is not Lipschitz continuous as shown below.

In $\mathbb{R}^2$, consider the triangle with two vertices fixed at $(\pm l, 0)$ and one moving vertex $(0, t\varepsilon)$ for $t \in [-1, 1]$. The signed area of the triangle changes from $-l\varepsilon$ (unbounded because $l$ can be large for any fixed small $\varepsilon$) to 0 (when $t = 0$ and the triangle degenerates), then to $l\varepsilon$ (when $t = 1$). The area changes by $2l\varepsilon$ while only one vertex moves by $2\varepsilon$, so the ratio of the area change over a point perturbation can be as large as a half-distance between given points.

**Lemma C.5** (time of strength). *For any base sequence $A$ of $n$ ordered points $p_1, \ldots, p_n \in \mathbb{R}^n$, the strength $\sigma(A)$ can be computed in time $O(n^3)$.*

**Proof of Lemma C.5.** The half-perimeter $p(A)$ is computable via all pairwise distances in time $O(n^2)$. The squared volume $\mathrm{vol}^2(A)$ can be expressed by the Cayley-Menger $(n + 2) \times (n + 2)$ determinant from Sippl & Scheraga (1986) in inter-point distances, which can be computed in time $O(n^3)$ by Lemma C.3. $\qquad\square$

**Lemma C.6** (axioms and time of $M_\infty$ on CBRs). *Let $G, F \subset \mathbb{R}^n$ be Euclidean graphs with $m$ unordered vertices and base sequences $A \subset V(G)$ and $B \subset V(F)$ of $h \leq n$ ordered vertices. The metric $M_\infty(\mathrm{CBR}(G; A), \mathrm{CBR}(F; B))$ from Definition 4.3 satisfies all metric axioms and is computable in time $O(m^{1.5} \log^{h+1} m)$ assuming that $n^3 \leq O(m^{1.5} \log^{h+1} m)$.*

**Proof of Lemma C.6.** The metric axioms for $M_\infty$ follow from the same axioms for the metrics $L_\infty$ and $W_\infty$ because the maximum of metrics is still a metric, see metric transforms in section 4.1 of Deza & Deza (2009). The first metric $\dfrac{2}{\lambda_n} |\mathrm{sign}(A)\sigma(A) - \mathrm{sign}(B)\sigma(B)|$ can be computed in time $O(n^3)$ by Lemma C.5. The metric $L_\infty(\mathrm{CD}(A), \mathrm{CD}(B))$ requires $O(h^2)$ time. The bottleneck distance $W_\infty(\mathrm{CR}(G; A)), \mathrm{CR}(F; B))$ between $(h + 1) \times (m - h)$ matrices $\mathrm{CR}(G; A)$, $\mathrm{CR}(F; B)$ with unordered columns (considered as clouds of $m - h$ unordered points in $\mathbb{R}^h$) needs time $O(m^{1.5} \log^{h+1} m)$ by Theorem 6.5 in Efrat et al. (2001). $\qquad\square$

**Lemma C.7** (metric axioms for NBM on NCDs). *The Nested Bottleneck Metric NBM from Definition 4.5 satisfies all metric axioms on Nested Distance Distributions.*

**Proof of Lemma C.7.** Induction on the depth $i = n, \ldots, 1$. The inductive base $i = n$ follows from the metric axioms in Lemma C.6 for $M_\infty$ in Definition 4.3.

The inductive step from a depth $i$ (between $1, n$) to the smaller value $i - 1$ follows from Lemma B.1 and the metric axioms in the inductive hypothesis for the depth $i$. $\qquad\square$

**Lemma C.8** (Lipschitz continuity of $M_\infty$). *Let $A$ be a base sequence of $1 \leq h \leq n$ ordered vertices in a Euclidean graph $G \subset \mathbb{R}^n$. Let $B, F$ be obtained from $A, G$, respectively, by perturbing every vertex of $G$ within its $\varepsilon$-neighborhood in $\mathbb{R}^n$. Then $\mathrm{CBR}(G; A)$ changes in $M_\infty$ from Definition 4.3 by at most $2\varepsilon$, so $M_\infty(\mathrm{CBR}(G; A), \mathrm{CBR}(F; B)) \leq 2\varepsilon$.*

**Proof of Lemma C.8.** Order all vertices of the graphs $G, F$ so that every vertex $p_i \in V(G)$ has the same index as its perturbation $q_i \in V(F)$. The bijection $p_i \leftrightarrow q_i$ induces the bijections between the corresponding elements of the matrices $\mathrm{CD}(A) \leftrightarrow \mathrm{CD}(B)$ and $\mathrm{CR}(G; A) \leftrightarrow \mathrm{CR}(F; B)$, which all differ by at most $2\varepsilon$. Lemma 4.2 implies that $\dfrac{2}{\lambda_n} |\mathrm{sign}(A)\sigma(A) - \mathrm{sign}(B)\sigma(B)| \leq 2\varepsilon$ Since all three components of the max metric $M_\infty$ in Definition 4.3 have the upper bound $2\varepsilon$, conclude that $M_\infty \leq 2\varepsilon$. $\qquad\square$

Definition B.2 can be adapted for the EMD between NCDs by (1) replacing the bottleneck distance $W_\infty$ in Definition 4.3 with EMD between clouds of equally weighted points, and (2) replacing $\mathrm{BMD}(\Gamma)$ for a bipartite graph $\Gamma$ with $\mathrm{EMD}(\Gamma)$ between the unordered sets (of potentially different sizes) of CBDs with weights on all white vertices and CBDs on all black vertices.

**Proof of Theorem 4.7(c).** We first prove the Lipschitz continuity of the metric NBM on NCDs. Order all vertices of the graphs $G, F$ so that every $p_i \in V(G)$ has the same index as its $\varepsilon$-perturbation $q_i \in V(F)$. In Definition 4.5, for any base sequence $A$ of $p_1, \ldots, p_h \in V(G)$, there is a base

sequence $B$ of vertices $q_1, \ldots, q_h \in V(F)$, which are $\varepsilon$-perturbations of $p_1, \ldots, p_h$, respectively, such that $M_\infty(\text{CBR}(G; A), \text{CBR}(F; B)) \leq 2\varepsilon$ by Lemma C.8.

These distances $M_\infty$ are weights of edges in the index-preserving vertex matching $E$ of the complete bipartite graph $\Gamma(G; p_1, \ldots, p_{h-1}; F; q_1, \ldots, q_{h-1})$ for any $p_1, \ldots, p_{h-1}$ and their $\varepsilon$-perturbations $q_1, \ldots, q_{h-1}$. Then $\text{BMD}(\Gamma(G; p_1, \ldots, p_{h-1}; F; q_1, \ldots, q_{h-1})) \leq 2\varepsilon$ by Lemma B.3. Since this conclusion holds for all (choices of) $p_1, \ldots, p_{h-1} \in V(G)$, we iteratively apply this argument for the bipartite graphs $\Gamma(G; p_1, \ldots, p_{i-1}; F; q_1, \ldots, q_{i-1})$ for $1 \leq i < n$ and finally conclude that $\text{NBM}(\text{NCD}(G; h), \text{NCD}(F; h)) \leq 2\varepsilon$. The proof that $\text{EMD}(\text{NCD}(G; h), \text{NCD}(F; h)) \leq 2\varepsilon$ is similar by using Lemma B.4 instead of B.3. $\square$

***Proof of Theorem** 4.7(d)*. In Definition 4.5, for any fixed $1 \leq i \leq h$ and ordered vertices $p_1 \ldots, p_{i-1} \in V(G)$ and $q_1 \ldots, q_{i-1} \in V(F)$, the bipartite graph $\Gamma(G; p_1, \ldots, p_{i-1}; F; q_1, \ldots, q_{i-1})$ has $V = 2(m - i + 1) = O(m)$ vertices, $E = (m - i + 1)^2 = O(m^2)$ edges.

For $i = h$, the weight $w(e)$ of each edge $e$ equals $M_\infty$, which needs time $O(m^{1.5} \log^{h+1} m)$ by Lemma C.6. For all $O(m^2)$ edges of $\Gamma(G; p_1, \ldots, p_{h-1}; F; q_1, \ldots, q_{h-1})$, the time is $O(m^{3.5} \log^{h+1} m)$. The bottleneck matching distance BMD for such a graph is computed by Hopcroft & Karp (1973) in time $O(E\sqrt{V}) = O(m^{2.5})$, which is dominated by the time $O(m^{3.5} \log^{h+1} m)$ preparing the weighted graph.

For all $O(m^{2(h-1)})$ choices of ordered vertices $p_1, \ldots, p_{h-1} \in V(G)$ and $q_1, \ldots, q_{h-1} \in V(F)$, the Bottleneck Matching Distance for all graphs $\Gamma(G; p_1, \ldots, p_{h-1}; F; q_1, \ldots, q_{h-1})$ are found in time

$$O(m^{2(h-1)})O(m^{3.5} \log^{h+1} m) = O(m^{2h+1.5} \log^{h+1} m).$$

For every next iteration $i = h - 2, \ldots, 1$, the parameter $i$ goes down by 1 every time. We can compute all distances $\text{BMD}(\Gamma(G; p_1, \ldots, p_{i-1}; F; q_1, \ldots, q_{i-1})$ in time

$$O(m^{2(i-1)})O(m^{3.5} \log^{h+1} m) = O(m^{2i+1.5} \log^{h+1} m).$$

The sum of all these times for $i = 1, \ldots, h - 1$ is still $O(m^{2h+1.5} \log^{h+1} m)$ from the first step.

All CBDs in Definition 3.5 have sizes at most $m$, which is the maximum number of points in the given clouds. The EMD between weighted distributions of a maximum size $m$ can be computed in near-cubic time $O(m^3 \log m)$, see Fredman & Tarjan (1987); Goldberg & Tarjan (1987). Since this complexity is dominated by the time $O(m^{3.5} \log^{h+1} m)$ for computing $O(m^2)$ weights $M_\infty$, each in time $O(m^{1.5} \log^{h+1} m)$ by Lemma C.6, the total time for the EMD is the same as for the NBM. $\square$

Thank you for reading all the proofs!

