# OpenReview forum: "Complete and Lipschitz continuous invariants of graphs under geometric isomorphism in R^n"
_ICLR.cc/2025/Conference — Submitted to ICLR 2025_

### Official Review · Reviewer_RkkB · 2024-10-30

**Soundness:** 3
**Presentation:** 2
**Contribution:** 3
**Rating:** 5
**Confidence:** 3

**Summary:**

In "Complete and Lipschitz continuous invariants of graphs under geometric isomorphism in R^n", the authors propose a Lipschitz-continuous representation of geometric graphs embedded into Euclidean space. The representation distinguishes all graphs which are not related by orientation-preserving isometries and is computable in polynomial time in the number of points m for a fixed dimension n. The authors achieve this via the introduced nested centred distribution, which is recursively constructed from representations ("Centred Distributions") for marked subsets of the point set.

**Strengths:**

* The paper introduces a provably complete and Lipschitz continuous invariant of geometric graphs.
* The invariant is computable in polynomial time for a fixed ambient dimension, which is the case for many geometric graphs.
* The proof for the Lipschitz continuity basically falls from the well-chosen definition.
* The first part of the paper contains many examples highlighting the strength of different invariants.

**Weaknesses:**

* There is very little intuition provided as to why this is the correct way to construct the invariant. In particular, an explanation of why we have to set n=h to get a complete invariant.
* Many things in this paper happen with the reader only understanding why they happen later. The roadmap for constructing our metric should be made much clearer already in the beginning. In the current state of the paper, we only learn on page 7 from Theorem 4.7 that the earlier introduced nested centred distribution will play the central role in this paper. This is mirrored in many other definitions, where the reader only understands later why they were useful definitions and presented at this particular point in the paper. I had to read the paper twice to understand it. (Admittedly, this is something Schopenhauer requested from readers of his book. I however don't believe that this is a good guiding principle for ICLR papers.)
* While the paper is technically solid in general, there are many minor technical issues in the definitions or the wording, see my minor comments below.
* It is not easy to follow the main part of the paper. In particular, the definitions of the nested bottleneck metric and the nested centred distribution provided are not, combined with the following notational problem: The definition of the centred distribution depends on the order h, i.e. CD_{h-1}(...) is defined differently than CD_{k-1}(...) for other k. However, this implicit order h does not appear in the notation. For example CD^h_{k} would solve this problem.
* There are over 8 different acronyms used throughout this paper. I understand that space does not permit to write these out every single time, but having them written out once in every definition they are used would greatly improve the readability of the paper. Furthermore, some important terms like R(G;A) and D(A) are missing from the lookup table 1.

**Questions:**

# Questions:

1. You claim that your NCD's have a vectorisation. Is this vectorisation unique and/or canonical? How can one compute this?
2. Are there any proofs of the minimal possible distance between two different geometric graphs?

# Minor comments:
* Line 26+: The definition of Euclidean graphs is a bit imprecise. You want your vertex set to be a subset of R^n, and this is basically all you need for a definition. Taking the geometric interpretation of edges into this makes it more complicated: Do you allow for overlapping edges? For a third vertex v to be contained in an edge? Do you consider G\subset R^n? What is actually the data of a geometric graph? Are the edges subsets of V^2, or of R^n?
* 59 up to 6 permutations -> up to all permutations
* 60 The space of triangular graphs is not the triangular cone, rather it can be represented by the triangular cone.
* Figure 2: "One isometry class of all isometric triangular graphs" -> this is not well defined, you mean the class of all triangular graphs isometric to a reference graph.
* 80 "Alternatively" -> Equivalently
* Problem 1:
	* There should be not whitespace in front of the colons.
	* You should define an abstract space S in which your invariant lives.
	* Then b) just reduces to stating that S is a metric space w.r.t. to a metric d and is Lipschitz continuous w.r.t to a certain metric on the space of geometric graphs.
	* Throughout the theorem, you use I to denote both: 1. the invariant mapping from the space of all geometric graphs to S and 2. a value I(G) in S on a geometric graph G.
	* I think that the notation G\subset R^n is confusing: there are multiple graphs geometric graphs which have the same underlying set in R^n.
* Line 94: DNA is not a good metaphor here. The geometry of proteins can vary although they are constructed from the same DNA.
* I don't think the meaning of line 98 is clear from the context.
* Line 103: You did not define unbounded graphs.
* Line 107: What means "preventing brute-force attempts" in this context? Maybe you could word that differently so I could understand it.
* Line 120 $Dot$->$\dot$
* I don't think the last sentence of the paragraph "Ordered clouds" is clear from the context.
* Line 127: The O(n) should be completely in the exponent.
* I don't think the last sentence of the paragraph "Unordered clouds" is clear from the context.
* Paragraph "Equivariants" Please introduce T_f
* The meaning of the first two sentences on page 4 are not clear at all to me.
* Line 175: I don't get the idea of the navigation metaphor.
* Definition 3.1:
	* I believe p and q become p_i and p_j later in the definition?
	* What does "has" mean in this context. Is it defined like this?
	* c) are we talking about the unordered rows OF D(G), or are the D(G)'s the rows themselves as stated in the definition.
* Line 197: What does local distribution mean?
* Line 207 and the rest of the paper: What is the motivation behind introducing PDD's and the other terms?
* Line 233: Please remind the reader of what the first k columns correspond to.
* Line 236: Counter-examples to what?
* Line 237: in the n\times n matrix -> in a n\times n matrix? Or about which matrix are we talking here? And the determinant has the sign sign(p_1,...,p_n), right?
* Definition 3.3: Please introduce p_0=0 more explicitly, this is easily overread. Furthermore, you reference to the points p_1, ...,p_n by "vectors", "vertices", and "points", which is confusing.
* Definition 3.5:
	* "and the center of mass" -> "with the center of mass"
	* As discussed above, you need to fix your notation and introduce the final order h to the formula for centred distributions CD_k^h
	* can you streamline this definition?
* Figure 3: Please mention in which direction the arrows face? Do the colours on the right have any meaning? I found the left part of the figure very hard to understand. (I only understood after I completely understood the definitions.) Can you add more explanation to this and make it more clear, which part of the diagram symbolises which part of your theory?
* Definition 4.1: You are confused about the dimensions of your objects: You start the definition by indexing points from 1 to n, but later switch to indexing points from 0 to n. I would assume that the n-dimensional simplex contains n+1 points, which seems to agree with the second part of the definition.
* Please introduce terms like p before you use them.
* Lemma 4.2: You should define epsilon and then make explicit on what the lambda_n depends. (Just on n, I would suppose)
* Line 315: If you define the metric L_\infty here, it should sound like a definition.
* Definition 4.3:
	* You write that your graphs "have" base sequences. A base sequence does not belong to the data of a geometric graph, hence wording this as "have" is confusing.
	* The term with the absolute value: This should be symmetric in A and B.
	* It is not clear whether this definition works. You want to define a metric between CR's, yet you use your knowledge of  A and B to compute this metric. You would need to prove that this does not depend on the choice of A and B, if they lead to the same CR.
* Line 324: Please specify the nodes of this graph.
* Definition 4.5: It would be great if this definition could be streamlined. The last part again suffers from the problem that you want to define a metric between NCD's, yet you access the points of the graphs used to construct the NCD's.
* Line 473: What part of this sentence is the heading?
* Line 521: There seems to be something wrong with this sentence.

---

> ### Author Response · Authors · 2024-11-17
> **Thank you for the detailed review**
>
> Thank you for the review.
>
> >the authors propose a Lipschitz-continuous representation of geometric graphs embedded into Euclidean space. The representation distinguishes all graphs which are not related by orientation-preserving isometries and is computable in polynomial time in the number of points m for a fixed dimension n. The authors achieve this via the introduced nested centred distribution, which is recursively constructed from representations ("Centred Distributions") for marked subsets of the point set.
>
> Thank you for the correct and detailed summary.
>
> >The paper introduces a provably complete and Lipschitz continuous invariant of geometric graphs. The invariant is computable in polynomial time for a fixed ambient dimension, which is the case for many geometric graphs. The first part of the paper contains many examples highlighting the strength of different invariants.
>
> Thank you for appreciating the effort of including many examples.
>
> >The proof for the Lipschitz continuity basically falls from the well-chosen definition.
>
> The Lipschitz continuity under rigid motion is highly non-trivial by covering all degenerate cases. For instance, when 3 points in the plane pass through a configuration within a straight line, the sign of orientation (of the triangle on 3 points) changes discontinuously, while the area (volume in higher dimensions) is non-Lipschitz.
>
> >There is very little intuition provided as to why this is the correct way to construct the invariant. In particular, an explanation of why we have to set n=h to get a complete invariant.
>
> Example 4.8 provides this intuition by proving the completeness in main Theorem 4.7(b) for n=1. Briefly, the choice h=n guarantees that the position of any vertex in R^n is uniquely defined by distances to the center of mass and h=n fixed vertices from a base sequence. The hardest part was to prove that these n+1 points still suffices to uniquely determine a collection (one of exponentially many) of all edges, not only the positions of vertices.
>
> >The roadmap for constructing our metric should be made much clearer already in the beginning. In the current state of the paper, we only learn on page 7 from Theorem 4.7 that the earlier introduced nested centred distribution will play the central role in this paper.
>
> Figure 3 right after Definition 3.4 of the nested centered distribution (NCD) highlights that NCD plays the central role as a complete invariant. The introduction only avoided forward references to specific definitions in later sections.
>
> >I had to read the paper twice to understand it. Admittedly, this is something Schopenhauer requested from readers of his book.
>
> Yes, re-reading is a good practice. If a paper is understood at first reading, improvements might be incremental. All big discoveries were different.
>
> >The definition of the centred distribution depends on the order h, i.e. CD_{h-1}(...) is defined differently than CD_{k-1}(...) for other k.
>
> Yes, as you say "for other k", more exactly for k<h, so there is no confusion.
>
> >this implicit order h does not appear in the notation. For example CD^h_{k} would solve this problem.
>
> Yes, we can include h as an extra parameter in the notation.
>
> >There are over 8 different acronyms used throughout this paper. I understand that space does not permit to write these out every single time, but having them written out once in every definition they are used would greatly improve the readability of the paper. Furthermore, some important terms like R(G;A) and D(A) are missing from the lookup table 1.
>
> Lookup Table 1 includes all important acronyms with references to their definitions. The distance matrix D(A) and matrix R(G;A) of relative distances are components of the Centered Representation CR(G;A) in Definition 3.3 references in Table 1.
>
> >You claim that your NCD's have a vectorisation. Is this vectorisation unique and/or canonical? How can one compute this?
>
> Lines 378-379: "Since every CR can be stored in a vector form, the complete invariant NCD(G; n) can be considered vectorial."
>
> Yes, if all Centered Representations are converted to vectors by flattening all matrices, the resulting collection can be considered canonical or unique, for example, by lexicographic ordering of vectors.
>
> >The definition of Euclidean graphs is a bit imprecise. You want your vertex set to be a subset of R^n, and this is basically all you need for a definition. Taking the geometric interpretation of edges into this makes it more complicated: Do you allow for overlapping edges? For a third vertex v to be contained in an edge? Do you consider G\subset R^n? What is actually the data of a geometric graph? Are the edges subsets of V^2, or of R^n?
>
> Th definition in lines 26-28 answered all these questions: "a Euclidean graph G in R^n is a finite set of m unordered (unlabeled) vertices located at distinct points of Rn and connected by straight-line edges."
>
> We continue below.

---

> > ### Author Response · Authors · 2024-11-17
> > **Continuation of the response above**
> >
> > >Do you consider G\subset R^n?
> >
> > Yes as written in line 26.
> >
> > >Do you allow for overlapping edges? For a third vertex v to be contained in an edge?
> >
> > Yes in both cases because no restrictions were specified.
> >
> > >What is actually the data of a geometric graph? Are the edges subsets of V^2, or of R^n?
> >
> > The input data of G consists of vertex positions and pairs of vertices for edges. That is why edges should be specified as straight-line segments connecting vertices. Missing the geometric interpretation of vertices would allow arbitrary continuous curves for edges as in topological graphs, see
> > https://en.wikipedia.org/wiki/Geometric_graph_theory.
> >
> > >"One isometry class of all isometric triangular graphs" -> this is not well defined, you mean the class of all triangular graphs isometric to a reference graph.
> >
> > This phrase was a short description within a picture, not a formal definition. The equivalence class is a collection of all objects equivalent to each other. There is no need to fix a reference object, see https://en.wikipedia.org/wiki/Equivalence_class.
> >
> > >Line 103: You did not define unbounded graphs.
> >
> > Here "unbounded" means "not contained within a fixed ball".
> >
> > >Line 107: What means "preventing brute-force attempts" in this context? Maybe you could word that differently so I could understand it.
> >
> > The full quote has the explanation: "Computability
> > 1.1(d) prevents brute-force attempts, e.g. defining I(G) as the infinite set of images of G under all rigid motions or taking m! distance matrices over all permutations of m unordered vertices."
> >
> > In other words, without computability condition 1.1(d), Problem 1.1 has the trivial solution: I(G) can be all infinitely many representations of G or the exponential-size set of all m! distances matrices obtained by all permutations of vertices.
> >
> > >I don't think the last sentence of the paragraph "Ordered clouds" is clear from the context.
> >
> > Lines 122-123: "an exponential time that can be close to O(2^m), not polynomial in the number m of ordered points as required in 1.1(d)."
> >
> > Is it clear that O(2^m) is not a polynomial complexity required in 1.1(d)?
> >
> > >I don't think the last sentence of the paragraph "Unordered clouds" is clear from the context.
> >
> > Lines 136-138: "one can choose any of m(m − 1)/2 edges and
> > produce 2^{m(m−1)/2} non-isometric graphs. Problem 1.1 for arbitrary graphs is computationally much harder than for point clouds due to exponentially many different graphs on the same vertex set."
> >
> > Is it clear that the number 2^{m(m−1)/2} potential collections of edges is exponential in the number m of vertices?
> >
> > >Paragraph "Equivariants" Please introduce T_f
> >
> > T_f is a transformation of equivariant values depending on f.
> >
> > >The meaning of the first two sentences on page 4 are not clear at all to me.
> >
> > Full quote: "Equivariants Gao et al. (2020); Qi & Luo (2020); Tu et al. (2022); Batzner et al. (2022) help predict forces acting on atoms to move them to a more optimal configuration. These time-dependent graphs G_t can be studied directly by invariant values I(Gt) without computing intermediate atomic forces".
> >
> > There is no need to spend time on predicting atomic forces by equivariants, because we can directly work with rigid classes of molecular graphs given by complete invariants.
> >
> > >Line 175: I don't get the idea of the navigation metaphor.
> >
> > Full quote: "Computing a metric between rigid classes of clouds is only a small part of Problem 1.1. Indeed, to efficiently navigate on Earth, in addition to distances between cities, we need a map of the whole planet and hence an invertible continuous invariant I, which is an analog of geographic coordinates."
> >
> > Is it clear that to go from a point A to a point B, it is not enough to know only the distance from A to B because finding a real path needs a geographic map, not only a distance value?
> >
> > >I believe p and q become p_i and p_j later in the definition?
> >
> > Yes, it was a general notation  for the Euclidean distance |p-q|.
> >
> > >What does "has" mean in this context. Is it defined like this?
> >
> > Yes, defined. We wrote "has" only to keep the definition in one line.
> >
> > >c) are we talking about the unordered rows OF D(G), or are the D(G)'s the rows themselves as stated in the definition.
> >
> > We meant "of D(G)", sorry that "of" was accidentally missed.
> >
> > >Line 197: What does local distribution mean?
> >
> > Another name for PDD as explained in lines 196-197: "The PDD was defined for finite clouds as a local distribution of distances in (M´emoli, 2011, Definition 5.5)".
> >
> > >Line 207 and the rest of the paper: What is the motivation behind introducing PDD's and the other terms?
> >
> > The motivation is in the title of section 3: build "a hierarchy of invariants from the fastest to complete".
> >
> > >Line 233: Please remind the reader of what the first k columns correspond to.
> >
> > This line in Example 3.2 follows Definition 3.1(c), where PDD was introduced with ordered columns by increasing distances to neighbors, hence the first k columns contain the first k shortest distances.

---

> > > ### Author Response · Authors · 2024-11-17
> > > **Continuation of the response above**
> > >
> > > >Line 236: Counter-examples to what?
> > >
> > > Full quote from lines 235-236: "Fig. S4 in Pozdnyakov et al. (2020) described infinitely many non-isometric pairs of clouds C,C′ in R^3 with PDD(C) = PDD(C′). These counter-examples inspired the stronger invariants for graphs"
> > >
> > > The equality PDD(C) = PDD(C′) means that PDD is incomplete, so these counter-examples are to the completeness of PDD.
> > >
> > > >Line 237: in the n\times n matrix -> in a n\times n matrix? Or about which matrix are we talking here?
> > >
> > > Full quote "Any n vectors p1, . . . , pn in Rn can be written as columns in the n × n matrix" defined *the* matrix whose columns are the vectors p1, . . . , pn.
> > >
> > > >the determinant has the sign sign(p_1,...,p_n), right?
> > >
> > > Yes, here "sign(p_1,...,p_n)" is the notation for the sign.
> > >
> > > >you reference to the points p_1, ...,p_n by "vectors", "vertices", and "points", which is confusing.
> > >
> > > It is the basic geometry: any vertex of G is a point of R^n and can be also interpreted as a vector.
> > >
> > > >Definition 3.5: can you streamline this definition?
> > >
> > > What is meant by "streamlining" here?
> > >
> > > >Figure 3: Please mention in which direction the arrows face?
> > >
> > > All arrows in Figure 3 have clearly specified directions.
> > >
> > > >Do the colours on the right have any meaning?
> > >
> > > Different invariants are shown in different colors.
> > >
> > > >I found the left part of the figure very hard to understand. (I only understood after I completely understood the definitions.)
> > >
> > > Yes, Figure 3 is right after Definition 3.5.
> > >
> > > >Can you add more explanation to this and make it more clear, which part of the diagram symbolises which part of your theory?
> > >
> > > The caption "Figure 3: Left: building the Nested Centered Distribution NCD from Definition 3.5 from Centered
> > > Representations in Definition 3.3 with metrics in section 4. Right: hierarchy of graph invariants."
> > >
> > > What exactly wasn't clear?
> > >
> > > >Definition 4.1: You are confused about the dimensions of your objects: You start the definition by indexing points from 1 to n, but later switch to indexing points from 0 to n.
> > >
> > > No confusion: the extra point p_0=0 denotes the center of mass, which is correctly higlighted in the notation sigma(A union 0).
> > >
> > > >I would assume that the n-dimensional simplex contains n+1 points, which seems to agree with the second part of the definition. Please introduce terms like p before you use them.
> > >
> > > Yes, the notation p was introduced by the keyword "let".
> > >
> > > >Lemma 4.2: You should define epsilon and then make explicit on what the lambda_n depends. (Just on n, I would suppose)
> > >
> > > Yes, we could the words "for any epsilon". The paramater is denoted by "lambda_n" showing the dependance only on n.
> > >
> > > >Line 315: If you define the metric L_\infty here, it should sound like a definition.
> > >
> > > Yes, the definition uses the keyword "is".
> > >
> > > >Definition 4.3: You write that your graphs "have" base sequences. A base sequence does not belong to the data of a geometric graph, hence wording this as "have" is confusing.
> > >
> > > The English meaning of the keyword "let" is the following: let A be a base sequence of a graph G.
> > >
> > > >The term with the absolute value: This should be symmetric in A and B.
> > >
> > > Sorry, this was a typo.
> > >
> > > >You want to define a metric between CR's, yet you use your knowledge of A and B to compute this metric. You would need to prove that this does not depend on the choice of A and B, if they lead to the same CR.
> > >
> > > The notation of the max metric M_inf(CR(G;A), CR(F;B)) clearly highlights the dependence on A,B. The first metric axiom for M_inf is proved in Lemma C.6, see lines 365-366: "The axioms of all metrics and main Theorem 4.7 below are proved in appendices B and C".
> > >
> > > >Line 324: Please specify the nodes of this graph.
> > >
> > > They are specified as "Centered Representations" in line 325: "bipartite graphs whose edge weights are the max metrics M_inf above on Centered Representations."
> > >
> > > >Definition 4.5: It would be great if this definition could be streamlined. The last part again suffers from the problem that you want to define a metric between NCD's, yet you access the points of the graphs used to construct the NCD's.
> > >
> > > We use only base sequences (point with indices up to h only) that are included in the invariant NCD.
> > >
> > > >Line 473: What part of this sentence is the heading?
> > >
> > > We can write with a verb: "Here is the main conclusion ..."
> > >
> > > >Line 521: There seems to be something wrong with this sentence.
> > >
> > > Is it clearer if we split this phrase into two?
> > >
> > > "The limitation is the time O(m^{2h+1.5} log^{h+1} m) of the metric NBM. This time is better than exponential and practical for h = 1, 2, 3 up to 29 atoms in QM9."
> > >
> > > If there are any more questions, please ask.

---

> > > > ### Comment · Reviewer_RkkB · 2024-11-22
> > > >
> > > > Thank you very much for the detailed answers!
> > > >
> > > > > The Lipschitz continuity under rigid motion is highly non-trivial by covering all degenerate cases. For instance, when 3 points in the plane pass through a configuration within a straight line, the sign of orientation (of the triangle on 3 points) changes discontinuously, while the area (volume in higher dimensions) is non-Lipschitz.
> > > >
> > > > Thank you for pointing out my misconception! I have updated my confidence score.
> > > >
> > > > In general, I feel that the paper presents a nice construction which I find interesting regardless of whether it can beat current methods on benchmark. As mentioned in my review and in the other reviews, in its current from the paper is hard to follow (at least for me) and provides not enough motivation and intuition for the constructions proposed (at least for me.) Your rebuttal suggests that you, as the authors, feel otherwise. I respect this, and want to emphasise that you know a lot more about the presented theory and the sub-field than I do.
> > > >
> > > > However from my point of view, only a revision of the paper providing clearer motivation and intuition and being easier to follow would warrant a raise of my score above a marginal reject. I do not think that this is a bad paper, but I also don't think that it is close to its best possible form.
> > > >
> > > > Many of my comments were intended to help you make the paper more precise and clearer. Of course, you are not obliged to follow up on them.
> > > >
> > > > I will respond to your questions below:
> > > >
> > > > >> I don't think the last sentence of the paragraph "Ordered clouds" is clear from the context.
> > > > >Lines 122-123: "an exponential time that can be close to O(2^m), not polynomial in the number m of ordered points as required in 1.1(d)."
> > > >
> > > > >Is it clear that O(2^m) is not a polynomial complexity required in 1.1(d)?
> > > >
> > > > I was referering to the sentence in lines 133--135
> > > > > Energy potentials written as infinite series of spherical harmonics, are often considered complete representations
> > > > of atomic environments, which holds in the limit but not for a finite sizePozdnyakov et al. (2020)
> > > >
> > > > > Th definition in lines 26-28 answered all these questions: "a Euclidean graph G in R^n is a finite set of m unordered (unlabeled) vertices located at distinct points of Rn and connected by straight-line edges."
> > > >
> > > > Marybe I am thinking a bit too formal here, but for me the data of the object is still not clear without guessing.
> > > >
> > > > > Lines 122-123: "an exponential time that can be close to O(2^m), not polynomial in the number m of ordered points as required in 1.1(d)."
> > > > >Is it clear that O(2^m) is not a polynomial complexity required in 1.1(d)?
> > > >
> > > > I was referring to "uses eigenvectors whose ambiguity up to signs gives an exponential time". I could guess what this could mean, but I would prefer not to guess.
> > > >
> > > > >The caption "Figure 3: Left: building the Nested Centered Distribution NCD from Definition 3.5 from Centered Representations in Definition 3.3 with metrics in section 4. Right: hierarchy of graph invariants."
> > > > >What exactly wasn't clear?
> > > > For me, it was not clear what the colours mean, where the part about the metric starts, whether the greed vertices in the metric part correspond to the green vertices on the left, in which direction the arrows point. Maybe it is just me being slow-to-comprehend, but this diagram was only clear to me me after I understood and thought about the definitions in the main text. Ideally, it should help me understand them.
> > > >
> > > > > "The limitation is the time O(m^{2h+1.5} log^{h+1} m) of the metric NBM. This time is better than exponential and practical for h = 1, 2, 3 up to 29 atoms in QM9."
> > > >
> > > > This is clearer to me, but I believe adding an "and" or an "for" or both in front of the "up to 29 atoms" would help as well.
> > > >
> > > > Thank you again for addressing my comments!

---

> > > > > ### Author Response · Authors · 2024-11-22
> > > > > **Thank you for the reply**
> > > > >
> > > > > >Thank you for pointing out my misconception! I have updated my confidence score.
> > > > >
> > > > > Because better understanding has been gained, could we ask why the confidence score was lowered from 4 to 3?
> > > > >
> > > > > >I feel that the paper presents a nice construction which I find interesting regardless of whether it can beat current methods on benchmark.
> > > > >
> > > > > Thank you for confirming this. The paper also reported experiments on past incomplete invariants. We emphasize that
> > > > > beating benchmarks on a finite dataset provides only numerical examples (almost always with failures below 100% accuracy), while rigorously proved Theorem 4.6 finishes Problem 1.1 for all possible embedded graphs in any R^n.
> > > > >
> > > > > >I was referering to the sentence in lines 133--135
> > > > >
> > > > > These lines are quoted below: "Energy potentials written as infinite series of spherical harmonics, are often considered complete representations of atomic environments, which holds in the limit but not for a finite size Pozdnyakov et al. (2020)."
> > > > >
> > > > > The series of spherical harmonics is infinite, not a finite size invariant.
> > > > >
> > > > > >>Lines 122-123: "an exponential time that can be close to O(2^m), not polynomial in the number m of ordered points as required in 1.1(d)." Is it clear that O(2^m) is not a polynomial complexity required in 1.1(d)?
> > > > >
> > > > > >Marybe I am thinking a bit too formal here, but for me the data of the object is still not clear without guessing.
> > > > >
> > > > > A physical object such as a molecular graph can be given in many data forms, e.g. coordinate representations, but all these different representations should be considered equivalent, so a rigid object can be formally defined as a rigid class of all such representations that can be exactly matched to each other under rigid motion.
> > > > >
> > > > > >I was referring to "uses eigenvectors whose ambiguity up to signs gives an exponential time".
> > > > >
> > > > > Lines 121-124: "multidimensional scaling known at least since 1935 Schoenberg (1935) can also provide an embedding C in R^k preserving all distances of C for a dimension k ≤ m. This embedding C in R^k uses eigenvectors whose ambiguity up to signs gives an exponential time that can be close to O(2^m), not polynomial in the number m of ordered points as required in 1.1(d)."
> > > > >
> > > > > A cloud of m unordered points with all inter-point distances 1 (the equlateral  (m-1)-dimensional simplex) requires an embedding into the space of dimension m-1, so k can be easily close to m. In this case, m-1 eigenvectors can be defined up to signs, which gives exponentially many 2^{m-1} choices. Moreover, in a case of symmetry (equal eigenvalues) eigenvectors are not uniquely defined at all.
> > > > >
> > > > > >>"The limitation is the time O(m^{2h+1.5} log^{h+1} m) of the metric NBM. This time is better than exponential and practical for h = 1, 2, 3 up to 29 atoms in QM9."
> > > > >
> > > > > >This is clearer to me, but I believe adding an "and" or an "for" or both in front of the "up to 29 atoms" would help as well.
> > > > >
> > > > > Yes, practical for h = 1, 2, 3 and for up to 29 atoms in QM9.
> > > > >
> > > > > If there are more questions, we can provide further clarifications.

---

> > > > > > ### Comment · Reviewer_RkkB · 2024-11-22
> > > > > >
> > > > > > Thanky you for your quick reply and your explanations!
> > > > > >
> > > > > > > Because better understanding has been gained, could we ask why the confidence score was lowered from 4 to 3?
> > > > > >
> > > > > > While I gained better understanding I realised at the same time that my understanding was worse than I initially thought. In order to not block the paper from getting accepted if other reviewers more knowledgeable suggest accpepting the paper, I lowered my confidence score.

---

### Official Review · Reviewer_Qm5F · 2024-10-31

**Soundness:** 2
**Presentation:** 1
**Contribution:** 2
**Rating:** 3
**Confidence:** 4

**Summary:**

The paper presents a hierarchy of invariants of Euclidean graphs embedded in $R^n$ under isometries in $R^n$. In particular, the authors present a novel invariant that is complete and Lipschitz-continuous (w.r.t. a metric constructed on the space of invariants). The invariant is thus stable w.r.t. small perturbations of the node positions. They demonstrate that the hierarchy of invariants can be used to distinguish all chemically different molecules in the QM9 dataset by treating the molecules as Euclidean graphs (discarding atom types).

**Strengths:**

* The posed problem and its individual aspects is stated clearly and motivated decently well.

* The proposed invariants seem to be technically valid and solve the posed problem (though I didn’t check any of the proofs in the appendix).

**Weaknesses:**

* The construction of the invariants higher up in the hierarchy CR, CD, NCD is not motivated well. Why was the CR triple chosen like this? Why does a nesting make sense? Why does one choose a base sequence? Why is a subset chosen up to the dimensionality of the embedding space? What is its influence?

* The definition of NCD is unclear, in particular l. 266-268 which seem to explain the nesting. Could there be a typo? How are the CD that are defined through CR (l.265) related to the CD that are defined through nesting l. 267? To clarify the construction, I would suggest to explain at the example of Fig.3 how to obtain NCD(G; 2).

* The motivation for the construction of the metric is also missing. Why does one consider the strength? Regarding the max metric $M_\infty$ (Def. 4.3), why does one take the maximum over the 3 different distance measures?

* The practical value of the novel invariant is unclear: The authors claim that it can be used as an informative embedding or as input for chemical property prediction (even tough the atom types are completely discarded). This is a strong statement an should be justified by a more expressive experiment than Fig. 6, e.g., considering more interesting chemical properties known for the QM9 dataset than number of atoms. If the invariant can only be used to discern molecules from different isometry classes, I wonder if not a simpler procedure suffices, e.g. a point cloud based approach plus a naive connectivity check.

* Formal problems in definitions:

  - The NCD is defined for $1 \le h \le n$ (Def. 3.5). but in example 3.6 $h=0$ is used.

  - Def. 4.1 of the strength requires $n$ points in $R^n$. However, in Def. 4.3 the strength is also used for base sequences of length $h < n$.

**Questions:**

* To construct an invariant for Euclidean graphs, why does it not suffice to use a distances-based invariant for point clouds with negative pairwise distances between non-adjacent nodes? Please elaborate further on the statement in l. 413-414.

* Are there no (perhaps not L-continuous) related methods to compare against?

* How does the presented approach relate to the following work: Nigam, Jigyasa, et al. "Completeness of atomic structure representations." _APL Machine Learning_ 2.1 (2024)?

* why does condition 1.1c) imply continuity as state in l.105?

* Can the invariant distinguish mirror images? I suspect not, since the construction is purely based on distances. If not, Problem 1.1 and other places which talk about “rigid motion” should be rephrased into related by $E(n)$ transformations (rigid motions and reflections).

* Is there a way to incorporate (invariant) node features like atom-types?

Minor points:

* I suspect a typo in def 3.4 (l.322) since the proposed metric is not symmetric in (G, A) and (F, B).

* definition where possible in formulae so there is less ambiguity. (like Def 4.3)

* l. 297 is unclear, I suspect what is meant is that the points are linearly dependent.
* l. 325 typo “above on”

* In Def 4.4 it seems to be missing that the collection of edges is chosen such that each vertex has degree 1 in order to guarantee a matching.

* I suggest not to use boxes at the end of definitions and examples (the boxes are typically reserved for proofs only).

* In l.244 use \ rather than minus to denote the difference of sets.

---

> ### Author Response · Authors · 2024-11-17
> **Thank you for the long review**
>
> >The paper presents a hierarchy of invariants of Euclidean graphs embedded in $R^n$ under isometries in $R^n$. In particular, the authors present a novel invariant that is complete and Lipschitz-continuous (w.r.t. a metric constructed on the space of invariants). The invariant is thus stable w.r.t. small perturbations of the node positions. They demonstrate that the hierarchy of invariants can be used to distinguish all chemically different molecules in the QM9
>
> Thank you for summarizing the main results. The completeness is under the stronger equivalence of rigid motion (orientation-preserving isometry).
>
> >The posed problem and its individual aspects is stated clearly and motivated decently well. The proposed invariants seem to be technically valid and solve the posed problem
>
> Thank you for confirming the solid technical solution.
>
> >The construction of the invariants higher up in the hierarchy CR, CD, NCD is not motivated well. Why was the CR triple chosen like this?
>
> The CR triple contains all required data for uniquely reconstructing any vertex and edge of G by starting from a base sequence.
>
> First, lines 117-119 reminded that "complete invariant of C under isometry (compositions of translations, rotations, reflections) is the classical m × m matrix Li et al. (2023) of pairwise distances |pi − pj |",which motivates using the distance matrix D(A) for a base sequence A.
>
> Second, the basic fact in Euclidean geometry says that any point in R^n is uniquely determined by n+1 distances to n+1 fixed points, which motivates the matrix R(G;A).
>
> Finally, the sign(A) is needed to distinguish mirror images.
>
> >Why does a nesting make sense?
>
> The nesting is needed to uniquely identify edges, which is explicitly used in the proof of Theorem 4.7(b) in lines 1035-1039:
>
> "we use the nested structure of NCD(G; n) to take another CBR(G; p1, . . . , pk, q, . . . , pn) for a new vertex q ∈ V (G) − A. Check if aff(p1, . . . , pk, q) = k + 1 again by Lemma C.1(b) using the matrix D(p1, . . . , pk, q)."
>
> Also in lines 1062-1066:
>
> "The nested structure of NCD(G; n) allows us to consider m − n unordered Base Representations CBR(G; p1, . . . , pn−1, pj) for all vertices pj with j = n, . . . ,m. Every vertex pj in V (G) is
> uniquely determined in Rn by the column of its signed distances to p0, . . . , pn in the (n+1)×(m−n−1) matrix R(G; p0, . . . , pn) for j = n + 1, . . . ,m."
>
> >Why does one choose a base sequence?
>
> To reconstruct any other vertex of G in R^n by using distances to fixed n+1 points: the center of mass and a base sequence.
>
> >Why is a subset chosen up to the dimensionality of the embedding space? What is its influence?
>
> Lines 1005-1006: "The proof of Theorem 4.7(b) will use the fact that any point in R^n is uniquely determined by n + 1
> distances to n + 1 ordered points that affinely span R^n".
>
> >The definition of NCD is unclear, in particular l. 266-268 which seem to explain the nesting. Could there be a typo?
>
> The only typo is the symbol C (instead of G) in line 267.
>
> >How are the CD that are defined through CR (l.265) related to the CD that are defined through nesting l. 267?
>
> All these Centered Distributions have different subscripts, which counts the number of points parameters, e.g. CD_{h-1}(G;p_1,...,p_{h-1}) in line 265, then CD_{k}(G;p_1,...,p_{k}) for another index k<h.
>
> >I would suggest to explain at the example of Fig.3 how to obtain NCD(G; 2).
>
> Yes, Figure 3 shows that NCD(G;2) of a triangular graph G consists of 3 Centered Representations CR(G;p_1,p_2), CR(G;p_2,p_3), CR(G;p_1,p_3).
>
> >The motivation for the construction of the metric is also missing. Why does one consider the strength?
>
> The strength is a Lipschitz continuous factor that smooths the discontinuous change of a sign (of orientation) when (say) 3 points in the plane go through a degenerate configuration within a straight line.
>
> >Regarding the max metric $M_\infty$ (Def. 4.3), why does one take the maximum over the 3 different distance measures?
>
> To guarantee the first metric axiom: the equality M_\infty = 0 should imply that all three components of Centered Representations. We need D(A)=D(B) to guarantee that the base sequences A and B can be matched by isometry. We need sign(A)=sign(B) to guarantee that the isometry matching A,B preserves orientation. We need R(G;A)=R(F;B) to guarntees that all other vertices of G,F have the same positions relative to the matched base sequences A,B.
>
> We continue below because of the space limit.

---

> > ### Author Response · Authors · 2024-11-17
> > **Continuation of the response above**
> >
> > >The authors claim that it can be used as an informative embedding or as input for chemical property prediction (even tough the atom types are completely discarded). This is a strong statement an should be justified by a more expressive experiment than Fig. 6
> >
> > The justified experiment is not in Figure 6 but was described in lines 116-122. The distances in Table 3 implied the main conclusions in lines 474-483:
> >
> > "complete invariants in Table 3. In other words, the map {molecules} -> {graphs on atomic centers (without chemical elements)} is injective on rigid classes. Hence this map can be theoretically inverted on its image, so we can reconstruct all chemistry from sufficiently precise geometry.
> >
> > The solution to Problem 1.1 settled the long-standing challenge of properly defining a molecular structure. A traditional approach including the recent one in Lang et al. (2023) is to describe such a structure as ‘a set of unlabeled configurations that are relatively similar to each other’. This ‘relative
> > similarity’ cannot depend on manual thresholds, which make any classification trivial as explained in Anosova et al. (2024), but should be rigorously defined as a class under equivalence (say, rigid motion) satisfying all axioms including the transitivity, which guarantees well-defined disjoint classes".
> >
> > >considering more interesting chemical properties known for the QM9 dataset than number of atoms.
> >
> > The most interesting property is the full molecular structure, now defined as a rigid class uniquely identified by the new complete invariants and hence determining all physical properties of a real molecule within the same environment such as temperature and pressure.
> >
> > >If the invariant can only be used to discern molecules from different isometry classes, I wonder if not a simpler procedure suffices, e.g. a point cloud based approach plus a naive connectivity check.
> >
> > Problem 1.1 goes far beyond an algorithmic detection, which were referenced in lines 124-130:
> >
> > Computational geometry developed many algorithms for detecting geometric isomorphism (or isometry, also called congruence) between point sets without edges Huttenlocher et al. (1993); Chew & Kedem (1992); Chew et al. (1999); Goodrich et al. (1999). (Arvind & Rattan, 2016, Theorem 3) describing, for a finite set A in Q^n of m points, n^O(n) poly(mM) time algorithm
> > to compute a canonizing function f(A), which can be considered a complete isometry invariant of A, where M upper bounds the binary encodings of the rational coordinates in the input".
> >
> > Any such detection gives a discontinuous metric, see lines 101-104: "one can define the discrete metric d(I(G), I(F)) = 1 for G not equivalent to F, which unhelpfully treats all non-equivalent graphs (even near-duplicates) as equally distant. The new requirement of Lipschitz continuity in 1.1(b) is much stronger than the classical ε−δ continuity because the constant λ is universal for all ε and unbounded graphs G in R^n."
> >
> > The Lipschitz continuity (especially under rigid motion distinguishing all mirror images) is the main advantage, not achieved by all past invariants of Euclidean graphs.
> >
> > If we missed any references, please quote the results that guarantee all the conditions in Problem 1.1.
> >
> > >The NCD is defined for $1 \le h \le n$ (Def. 3.5). but in example 3.6 $h=0$ is used.
> >
> > The case h=0 is exceptional. Example 3.6 discusses the simplest analog of NCD with an empty base sequence for h=0.
> >
> > >Def. 4.1 of the strength requires $n$ points in $R^n$. However, in Def. 4.3 the strength is also used for base sequences of length $h < n$.
> >
> > The last line in Def 4.3 "where all signs are zeros for h < n" implies that the strengths are irrelevant for h<n.
> >
> > >why does it not suffice to use a distances-based invariant for point clouds with negative pairwise distances between non-adjacent nodes? Please elaborate further on the statement in l. 413-414.
> >
> > >Are there no (perhaps not L-continuous) related methods to compare against?
> >
> > Because atoms always vibrate around their average positions, see Richard Feynman's first lecture on physics (Atoms in motion), any non-Lipschitz methods are hard to justify.
> >
> > >why does condition 1.1c) imply continuity as state in l.105?
> >
> > Full quote: "Condition 1.1(c) requires I to be not only complete and continuous but also efficient to explicitly
> > reconstruct G".
> >
> > Hence line 105 does not claim that 1.1(c) implies continuity. Condition 1.1(c) is about invertibility, which is an extra property in addition to the completeness and Lipschitz continuity in conditions 1.1(a,b).
> >
> > >I suspect a typo in def 3.4 (l.322) since the proposed metric is not symmetric in (G, A) and (F, B).
> >
> > Yes, thank for spotting this typo.
> >
> > >definition where possible in formulae so there is less ambiguity. (like Def 4.3)
> >
> > What is meant by "definition where possible in formulae"?
> >
> > We continue below because of the space limit.

---

> > > ### Author Response · Authors · 2024-11-17
> > > **Continuation of the response above**
> > >
> > > >How does the presented approach relate to the following work: Nigam, Jigyasa, et al. "Completeness of atomic structure representations." APL Machine Learning 2.1 (2024)"?
> > >
> > > This paper studies point clouds (bounded atomic environments), not graphs with edges but we can add this references if you wish.
> > >
> > > >Can the invariant distinguish mirror images? I suspect not, since the construction is purely based on distances.
> > >
> > > Line 370: Theorem 4.7(b) "NCD(G; n) is a complete invariant of graphs G in R^n under rigid motion from the group SE(n)."
> > >
> > > The constructions in Definitions 3.3 and 3.4 explicitly include signs that distinguish mirror images. In fact, the Lipschitz continuity under rigid motion is harder than under isometry, and requires the non-trivial concept of a strength of a simplex.
> > >
> > > Moreover, the bottom right image in Figure 4 shows the near mirror images that are still distinguished by the new invariant as explained in the caption.
> > >
> > > >Is there a way to incorporate (invariant) node features like atom-types?
> > >
> > > Lines 526-528: "Any vertex p and edge of G can have an attribute and a weight respected by any isometry that maps one graph to another. These vertex attributes and edge weights can be incorporated as extra columns and rows in CRs from Definition 3.3, and then incorporated into NCD and NBM."
> > >
> > > >l. 297 is unclear, I suspect what is meant is that the points are linearly dependent.
> > >
> > > Full quote: "When a base sequence A = (p1, . . . , pn) in R^n degenerates to a lower dimensional subspace"
> > >
> > > In other words, when points p1, . . . , pn become linearly dependent.
> > >
> > > >l. 325 typo “above on”
> > >
> > > There is no typo, but the word "above" can be skipped.
> > >
> > > >In Def 4.4 it seems to be missing that the collection of edges is chosen such that each vertex has degree 1 in order to guarantee a matching.
> > >
> > > Full quote: "Let Γ be a complete bipartite graph with m white vertices and m black vertices so that every white vertex is connected to every black vertex by a single edge.
> > >
> > > The keyword *single* guarantees a 1-1 bijective matching.
> > >
> > > >I suggest not to use boxes at the end of definitions and examples (the boxes are typically reserved for proofs only).
> > >
> > > Than you for the suggestion. Proofs are finished with white boxes, not blac boxes.
> > >
> > > >In l.244 use \ rather than minus to denote the difference of sets.
> > >
> > > Ok. We hope that all questions have been answered.
> > >
> > > However, if anything remains unclear, we can clarify.

---

> ### Comment · Reviewer_Qm5F · 2024-11-21
> **Thank you for the response**
>
> The construction presented in the current draft lacks in many places motivation (see many questions above: motivation of the strenght, the metric, the role of the base sequence, role of nesting,.. ). In several places the construction is unclear and inaccurate. The experiments do not sufficiently convey the practical value of the invariant.
> Addressing the questions and concerns raise above requires a major revision of the current draft. I therefore do not find the presented work ready for publication at ICLR.
>
> **Open points**:
> Regarding the practical value of the invariant: the authors have defended their complete invariants as a map of molecules. However, to me the usefulness of this map is yet to be proven.
> The authors show it can be used to distinguish molecules under their metric, but could it be used as an informative embedding, that is, does it carry/summarize (domain specific) information in a practically usable format?
>
> Regarding the question for a comparison to other methods: the author say that non-Lipschitz methods are hard to justify but if the authors are aware of methods that are applicable (perhaps to QM9 molecules as point clouds discarding the edges), it would strengthen their work a lot if they would compare favorably against existing methods. Could they for instance compare against Nigam, Jigyasa, et al. "Completeness of atomic structure representations." APL Machine Learning 2.1 (2024)"?
> On a more subtle note: chemical bonds are derived from atom positions and electron densities. So, to construct a complete map of chemistry starting from the atom types and atom positions should suffice.
>
> **Minor points**:
> Regarding the suggstion to use equations for mathematical definitions where possible, I meant the following: e.g. in Def 4.3 use
>
> $$M_\infty(CR(G; A), CR(F;B)) := \max(\frac{2}{\lambda} …, \text{2nd term}, \text{3rd term})$$
>
> Instead of saying this in words. It helps, for instance, when trying to spot a definition quickly and is less prone to misunderstandings.

---

> ### Author Response · Authors · 2024-11-21
> **Thank you for the reply**
>
> >many questions above: motivation of the strenght, the metric, the role of the base sequence, role of nesting,
>
> These are motivations for technical constructions. Because the short conference version cannot include motivations for all such details, we motivated the main concepts such as a complete invariant and and a distance metric (satisfying all metric axioms) in Problem 1.1.
>
> >In several places the construction is unclear and inaccurate.
>
> If you could specify line numbers, we can provide further clarifications.
>
> > Regarding the practical value of the invariant: the authors have defended their complete invariants as a map of molecules.
>
> The main contribution (Theorem 4.7 fully solving Problem 1.1) stands on its own even without any experiments, because all past invariants could not guarantee complete invariants with Lipschitz continuous metrics, all computable in polynomial time for a fixed dimension.
>
> >The authors show it can be used to distinguish molecules under their metric, but could it be used as an informative embedding, that is, does it carry/summarize (domain specific) information in a practically usable format?
>
> Thank you for proposing the important questions for further developments. The current paper is already too long with appendices.
>
> >non-Lipschitz methods are hard to justify but if the authors are aware of methods that are applicable (perhaps to QM9 molecules as point clouds discarding the edges), it would strengthen their work a lot if they would compare favorably against existing methods
>
> The most important comparison should be by theoretical guarantees, not by experimental studies. Running extra experiments on discontinuous or incomplete descriptors would waste time and resources without proving anything because numerical examples can only disprove conjectures but cannot prove claims that should hold for all infinitely many objects.
>
> >Could they for instance compare against Nigam, Jigyasa, et al. "Completeness of atomic structure representations." APL Machine Learning 2.1 (2024)"?
>
> We can cite this paper, which studies point clouds (atomic environments) without edges. However, do you agree that any invariants of point clouds cannot distinguish exponentially many embedded graphs with the same set of vertices? The experiments also compared the new invariants with other faster but incomplete invariants. May we ask why you are proposing this paper again?
>
> >chemical bonds are derived from atom positions and electron densities. So, to construct a complete map of chemistry starting from the atom types and atom positions should suffice.
>
> Yes, we study point clouds in another work. Chemical bonds have no precise definitions and depend on many manually chosen thresholds for distances and angles. Theorem 4.6 applies to all embedded graphs in any dimension, not only to molecular graphs.
>
> >Regarding the suggstion to use equations for mathematical definitions where possible, I meant the following: e.g. in Def 4.3 use
> $$M_\infty(CR(G; A), CR(F;B)) := \max(\frac{2}{\lambda} …, \text{2nd term}, \text{3rd term})$$ Instead of saying this in words.
>
> We tried this when writing the paper. Unfortunately, the formula is a bit too long and cannot fit one line in the ICLR template.
>
> >It helps, for instance, when trying to spot a definition quickly and is less prone to misunderstandings.
>
> Other people might prefer more words than symbols, so it is only a style.

---

### Official Review · Reviewer_N6Dg · 2024-11-04

**Soundness:** 3
**Presentation:** 2
**Contribution:** 3
**Rating:** 8
**Confidence:** 2

**Summary:**

This paper proposes a new class of invariants for graphs that maintain desirable geometric properties that are defined algebraically, but additionally possess the property of Lipschitz continuity which is a form of stability that is useful when considering data arising from a random data generating process.

**Strengths:**

Overall, I found this paper to be a strong contribution though I am admittedly not very confident in some of the technical areas that this paper covers, mostly in terms of existing work and their limitations, though the background and literature appears to be complete.

The paper appears to be technically solid and is mathematically interesting.

The experiments were good, it was interesting to see the performance and scalability of the approach on a real-life relevant dataset.

Good overview on the limitations of the work, including future directions of research that can be carried to overcome them.

**Weaknesses:**

I have no familiarity nor experience with chemical data, so this concern may not make sense, but it appears to me that molecule structure is something quite well-defined with a clear construction where randomness is probably not that important (it could be rather naïve thinking, but if the conditions aren't right for a molecule to form, then it won't form so there needs to be some inherent stability to the setting anyway, so the contribution of Lipschitz continuity as an important property doesn't seem to be practically that relevant here).  So, while the results look quite impressive in the experimental application, it's not immediately obvious to me that this is an interesting application area to demonstrate the result.

The paper could be better written, the writing style feels colloquial to me (contractions typically aren't used in formal papers).  The style is also less typical of computer science conference papers where there is typically a clear bullet point list of the contributions of the work.

**Questions:**

- I may have missed it, but would it be possible to have a concrete experimental example/experimental comparison against existing work and what breaks down if the Lipschitz contribution that the work is proposing doesn't hold?  This would better illustrate why this result is needed.
- Is there any intuition on how the contribution benefits other application areas other than molecule classification?
- It is briefly mentioned at the end of the paper that the work might extend to more general cases with other data structures, however, it would be interesting to understand the current setting but under more complex assumptions, such as graphical models.  Is there any intuition available on how to develop and adapt the results under of nodes/edges (e.g., how would the constant change)?  Could the results contribute to the area of causal inference?

---

> ### Author Response · Authors · 2024-11-16
> **Thank you for understanding the significance**
>
> Sorry about the slow reply caused by earlier commitments this week.
>
> >This paper proposes a new class of invariants for graphs that maintain desirable geometric properties that are defined algebraically, but additionally possess the property of Lipschitz continuity which is a form of stability that is useful when considering data arising from a random data generating process.
>
> Thank you for correctly summarizing the contributions
>
> >The paper appears to be technically solid and is mathematically interesting. The experiments were good, it was interesting to see the performance and scalability of the approach on a real-life relevant dataset. Good overview on the limitations of the work, including future directions of research that can be carried to overcome them.
>
> Thank you for highlighting the key strengths including solid mathematical results and scalability on a real-life dataset.
>
> >it appears to me that molecule structure is something quite well-defined with a clear construction where randomness is probably not that important (it could be rather naïve thinking, but if the conditions aren't right for a molecule to form, then it won't form so there needs to be some inherent stability to the setting anyway, so the contribution of Lipschitz continuity as an important property doesn't seem to be practically that relevant here).
>
> Richard Feynman's first lecture on physics (Atoms in motion) highlights that all atoms vibrate around their average positions. Hence any experimentally determined atomic structure includes bounded noise (also due to the uncertainty of measurements), which motivates the search for Lipschitz continuous invariants of molecular graphs under perturbations of vertices.
>
> >while the results look quite impressive in the experimental application, it's not immediately obvious to me that this is an interesting application area to demonstrate the result.
>
> Any iterative energy optimization of molecular structures inevitably stops at some approximation to a local optimum. Hence such molecular simulations can easily generate thousands of near-duplicate molecules around the same optimum. This duplication requires quick filtering by Lipschitz continuous invariants.
>
> >the writing style feels colloquial to me (contractions typically aren't used in formal papers).
>
> Yes, we can remove all contractions.
>
> >The style is also less typical of computer science conference papers where there is typically a clear bullet point list of the contributions of the work.
>
> This style does not seem to be prescribed by the call of papers at https://www.iclr.cc/Conferences/2025/CallForPapers
>
> >would it be possible to have a concrete experimental example/experimental comparison against existing work
>
> Yes, Figure 4 has several important images: the left-hand side image compares the new metric NBM on complete invariants NCD with the EMD metric on the past incomplete invariants PDD, which cannot distinguish any mirror images; the bottom right image shows the specific molecules in QM9, which are (near) mirror images and 100x times better distinguished by NCD.
>
> >what breaks down if the Lipschitz contribution that the work is proposing doesn't hold? This would better illustrate why this result is needed.
>
> If an invariant discontinuously changes under a tiny perturbation of atoms, any machine learning on such unstable outputs becomes unreliable because gradient-based methods assume the even stricter smoothness, which implies continuity, see https://en.wikipedia.org/wiki/Smoothness.
>
> >Is there any intuition on how the contribution benefits other application areas other than molecule classification?
>
> Problem 1.1 is stated and solved for any embedded straight-line graphs in R^n and hence also covers geometric skeletons of any macroscopic physical objects or polygonal meshes of surfaces.
>
> >It is briefly mentioned at the end of the paper that the work might extend to more general cases with other data structures, however, it would be interesting to understand the current setting but under more complex assumptions, such as graphical models. Is there any intuition available on how to develop and adapt the results under of nodes/edges (e.g., how would the constant change)? Could the results contribute to the area of causal inference?
>
> Yes, embedded graphs (objects) and rigid motion (equivalence) in Problem 1.1 can be replaced with any other data objects and equivalences, respectively, while the problem of finding complete invariants with Lipschitz continuous metrics, all computable in polynomial time, remains the same. The first crucial step is to define a practically important equivalence relation on new objects.
>
> If you could clarify the meaning of "adapt the results under of nodes/edges", we would be happy to explain more details.

---

### Official Review · Reviewer_t58o · 2024-11-05

**Soundness:** 2
**Presentation:** 2
**Contribution:** 1
**Rating:** 1
**Confidence:** 5

**Summary:**

This paper studies the problem of producing an invariant of Euclidean graphs that gives rise to a metric but moreover is complete in the sense that the graph can be reconstructed from the invariant.  The desired properties of this invariant are characterized axiomatically in a fashion that ensures that the metric is useful for distinguishing graphs and computable.  After a brief review of some of the existing literature on this subject, the paper defines a new invariant (which is related to the "distance distribution" of each vertex) and studies its properties.  A numerical experiment on molecules from the "QM9" database validates the theoretical results.

**Strengths:**

The paper is technically sound.

**Weaknesses:**

The paper has a number of weaknesses:

1) It is very poorly written, and in particular the exposition is both bombastic and also lacks clarity in various places.

2) The comparison to existing methods (e.g., the Gromov-Hausdorff or Gromov-Wasserstein distances) is remarkably ungenerous; I do not think a reasonable assessment of the method is given.

3) It is not really motivated why a complete invariant is needed as opposed to just a metric.

4) The evaluation is limited; the importance of the QM9 database is not explained, nor is there any comparison to any other method for solving this problem.  And there are no real applications other than distinguishing molecules.

The problem of finding distances between metric spaces or point clouds or distributions is one that has a long history and for which there are many algorithms in the literature.  This paper does not seriously motivate the contribution it purports to make, nor does it engage with this literature in an honest way.

**Questions:**

1) Why do we want a complete invariant?

2) Why is this method better than, say, approximating the Gromov-Wasserstein distance using the new efficient approximation methods?  How do other methods perform on the data set used for evaluation?

3) What is the real application of such a metric on Euclidean graphs?  Distinguishing molecules sort of begs the question.

---

> ### Author Response · Authors · 2024-11-17
> **answers and questions**
>
> >This paper studies the problem of producing an invariant of Euclidean graphs that gives rise to a metric but moreover is complete in the sense that the graph can be reconstructed from the invariant. The desired properties of this invariant are characterized axiomatically in a fashion that ensures that the metric is useful for distinguishing graphs and computable.
>
> Thank you for highlighting the main results.
>
> >I am not enthusiastic about this paper
>
> Your feelings are acknowledged. How does this phrase follow the official guidance below?
>
> "Strengths: A substantive assessment of the strengths of the paper, touching on each of the following dimensions: originality, quality, clarity, and significance. We encourage reviewers to be broad in their definitions of originality and significance. For example, originality may arise from a new definition or problem formulation, creative combinations of existing ideas, application to a new domain, or removing limitations from prior results. You can incorporate Markdown and Latex into your review. See https://openreview.net/faq."
>
> > am hard-pressed to find substantial strengths.
>
> May we ask who has "hard-pressed" you? If this "hard-pressing" is against your will, should ethical concerns be raised?
>
> In fact, the strengths were highlighted in the above summary, quoted below:
>
> "an invariant of Euclidean graphs that gives rise to a metric but moreover is complete in the sense that the graph can be reconstructed from the invariant"
>
> "properties of this invariant are characterized axiomatically in a fashion that ensures that the metric is useful for distinguishing graphs and computable"
>
> >the exposition is both bombastic
>
> We were shocked to find a military-style language in a supposedly scientific review, especially because many people really die in several on-going wars across the world right now.
>
> Could you please include the quotes that were called "bombastic"?
>
> >and also lacks clarity in various places.
>
> Could you please specify the line numbers that might need clarifications?
>
> >The comparison to existing methods (e.g., the Gromov-Hausdorff or Gromov-Wasserstein distances) is remarkably ungenerous; I do not think a reasonable assessment of the method is given.
>
> These distances are reviewed in lines 169-174, quoted below:
>
> "Gromov-Wasserstein metrics M´emoli (2011) are defined for any metric-measure spaces Br´echeteau (2019) by minimizing over infinitely many correspondences between points, but cannot be approximated with a factor less than 3 in polynomial time unless P=NP (Schmiedl, 2017, Corollary 3.8) and polynomial algorithms for partial cases in M´emoli et al. (2021); Majhi et al. (2024). The related problems of matching and finding distances between fixed Euclidean graphs (but not for their isometry classes) were studied in Nikolentzos et al. (2017); Majhi & Wenk (2022); Buchin et al. (2023)."
>
> The very first phrase generously highlighted that these metrics "are defined for any metric-measure spaces", which is much more general than the case of Euclidean graphs in R^n.
>
> Could we clarify what phrases were found "remarkably ungenerous"?
>
> If we accidentally missed any references, could you please specify them, especially with time complexities?
>
> >It is not really motivated why a complete invariant is needed as opposed to just a metric.
>
> Lines 175-177: "Computing a metric between rigid classes of clouds is only a small part of Problem 1.1. Indeed, to
> efficiently navigate on Earth, in addition to distances between cities, we need a map of the whole planet and hence an invertible continuous invariant I, which is an analog of geographic coordinates".
>
> Is it clear that to go from a point A to a point B (needed to explain phase transitions or continuous paths in molecular dynamics), it is not enough to know only the distance from A to B because finding a real path needs a geographic map, not only a distance value?
>
> >The evaluation is limited; the importance of the QM9 database is not explained,
>
> QM9 is the well-known public database containing 130K+ molecules with atomic coordinates, while most other databases have only chemical compositions or abstract graph information (chemical bonds) without real coordinates. Another well-known database PDB (Protein Data Bank) is not relevant in our case of unordered atoms because all atoms in a protein chain are ordered along a protein backbone.
>
> >nor is there any comparison to any other method for solving this problem.
>
> The comparisons with the past incomplete invariants SRV, SDV, PDD are described in lines 416-422, Table 3, and Figure 4.
>
> We continue below because of the space limit.

---

> ### Author Response · Authors · 2024-11-17
> **Continuation of the response above**
>
> >there are no real applications other than distinguishing molecules.
>
> Problem 1.1 is stated and solved for any embedded straight-line graphs in R^n and hence also covers geometric skeletons of any macroscopic physical objects or polygonal meshes of surfaces.
>
> >The problem of finding distances between metric spaces or point clouds or distributions is one that has a long history and for which there are many algorithms in the literature. This paper does not seriously motivate the contribution it purports to make, nor does it engage with this literature in an honest way.
>
> To justify these claims, could you please quote specific theorems in the past literature that solve Problem 1.1 by satisfying all the conditions (a), (b), (c), (d)?
>
> If we missed any relevant references, we would be happy to insert them.
>
> >Why do we want a complete invariant?
>
> Not only a complete but also an invertible invariant as in condition 1.1(c) is needed to explain structure-property relations by predicting (better expressing) important physical properties in terms of complete invariants. For example, the life-critical data (including average annual temperature, precipitation etc) of any place on Earth essentially require geographic coordinates (complete and invertible invariants), not only pairwise distances between (infinitely many) locations.
>
> >Why is this method better than, say, approximating the Gromov-Wasserstein distance using the new efficient approximation methods?
>
> First, do you agree that the Gromov-Wasserstein distance "cannot be approximated with a factor less than 3 in polynomial time unless P=NP (Schmiedl, 2017, Corollary 3.8)"?
>
> If you have any conflict of interest with us quoting this result above, we would appreciate more justifications.
>
> Second, we mentioned polynomial algorithms in several partial cases below.
>
> Lines 172-174: "polynomial algorithms for partial cases in M´emoli et al. (2021); Majhi et al. (2024). The related problems of matching and finding distances between fixed Euclidean graphs (but not for their isometry classes) were studied in Nikolentzos et al. (2017); Majhi & Wenk (2022); Buchin et al. (2023)."
>
> If you would like us to add more references, please specify them.
>
> Third, lines 175 underscored that "Computing a metric between rigid classes of clouds is only a small part of Problem 1.1."
>
> Moreover, solving Problem 1.1 does not cancel important efforts in improving other metrics or invariants, so feel free to develop "new efficient approximation methods" for Gromov-Wasserstein distance or any other distances as you like.
>
> If you claim that Gromov-Wasserstein distances satisfy the Lipschitz continuity and polynomial-time computability as in Problem 1.1, please quote exact references. Scientific integrity requires not to block advances that rigorously solve other problems, see https://iclr.cc/public/CodeOfEthics
>
> Could you please confirm your commitment to the highest level of scientific integrity?
>
> >How do other methods perform on the data set used for evaluation?
>
> All past distance-based invariants cannot distinguish mirror images, see the specific examples in the bottom right image of Figure 4. If you know any Lipschitz continuous invariants that can distinguish mirror images of Euclidean graphs, please quote the relevant results.
>
> >What is the real application of such a metric on Euclidean graphs? Distinguishing molecules sort of begs the question.
>
> The applications are described in lines 475-483 quoted below:
>
> "the map {molecules} -> {graphs on atomic centers (without chemical elements)} is injective on rigid classes. Hence this map can be theoretically inverted on its image, so we can reconstruct all chemistry from sufficiently precise geometry.
>
> The solution to Problem 1.1 settled the long-standing challenge of properly defining a molecular structure. A traditional approach including the recent one in Lang et al. (2023) is to describe such a structure as ‘a set of unlabeled configurations that are relatively similar to each other’. This ‘relative similarity’ cannot depend on manual thresholds, which make any classification trivial as explained in Anosova et al. (2024), but should be rigorously defined as a class under equivalence (say, rigid motion) satisfying all axioms including the transitivity, which guarantees well-defined disjoint classes."
>
> What exactly did you mean by "begs the question"?
>
> Could you please justify the score 1 (poor) according to the official guidance below?
>
> "Contribution* Please assign the paper a numerical rating on the following scale to indicate the quality of the overall contribution this paper makes to the research area being studied. Are the questions being asked important? Does the paper bring a significant originality of ideas and/or execution? Are the results valuable to share with the broader ICLR community?"
>
> Looking forward to receiving responses to the questions above.

---

> > ### Comment · Reviewer_t58o · 2024-11-22
> > **Reply and elaboration**
> >
> > Let me try to be more specific about some of my criticism and concerns about the paper.
> >
> > 1) The authors talk about their construction as giving coordinates on the space of graphs/ molecules.  Typically this means producing a map from the data to some Euclidean space R^n, or at the very least to some kind of manifold.  However, the definition 1.1 is vague about where I is supposed to take values.  Moreover, the actual construction of NCD in 4.5 does not seem to take values in a vector space or anything like it but rather is defined in terms of a collection of unordered sets --- although it is not at all clear from the definition what the target metric space is and one has to untangle a series of prior definitions to understand it.
> >
> > This seems both like a conceptual issue -- in what sense are these coordinates, beyond the fact that they live in a metric space, and also an expository one, since both the motivating question and the definitions are not clear on this point.
> >
> > 2) The problem that the authors are solving is in fact solved by (variants of) the Gromov-Hausdorff distance; it is very easy to see that it satisfies the Lipshitz condition, except for the question of polynomial-time computability.  The authors are correct that it is known to be NP-hard to approximate in general.  But on the other hand, as far as I can tell the invariant the authors introduce is basically computationally infeasible for data sets larger than maybe 100 points or so (please correct me if this is false), and in particular for the application in question the largest molecules seem to be on the order of maybe 30 points.  It is well-known that Gromov-Hausdorff can be computed for such small point clouds.  Moreover, the entropically-regularized Gromov-Wasserstein approximations can efficiently handle data sets with thousands of points.  The authors do not seriously compare to this kind of approach nor do they give a real sense of how large a data set the NCD invariant can handle.
> >
> > 3) The fact that something like the NCD invariant characterizes finite metric spaces is closely related to Gromov's theorem showing that metric measure spaces are uniquely characterized by the distributions induced on finite distance matrices.  Although the authors briefly mention metric geometry, they do not mention this nor any of the work in this direction (e.g., Greven-Pfaffelhuber-Winter).
> >
> > 4) The authors are very emphatic about the importance of their invariant on the data set QM9, but no indication is given of the scientific value of being able to characterize this data set in this fashion.  I read the references appealed to, and I still cannot find an answer to the question of what scientific problem has been solved.  The problem of "finding coordinates" requires a why -- for example, in genomics finding coordinates is then often used to cluster and do clinical inference on the effect of treatment.  What is the basic question here?  Why was it not handled by the approximations people tried before?
> >
> > In light of these issues, I remain firm that I do not think this paper is suitable for publication.  Having read the criteria for the ratings, I am willing to update my rating to a 2, as there are no ethical issues and the result is not completely trivial, however.

---

> > > ### Author Response · Authors · 2024-11-22
> > > **Thank you for the reply**
> > >
> > > >The authors talk about their construction as giving coordinates on the space of graphs/ molecules. Typically this means producing a map from the data to some Euclidean space R^n, or at the very least to some kind of manifold.
> > >
> > > Yes, all components of the new invariant NCD are real numbers. One can embed the space of NCD invariants into some Euclidean space and then apply identifications. However, to work with real data, this high-dimensional embedding is not needed at all. In the geographic analogy, most maps easily with two coordinates (latitude and longitude), whose ranges define a rectangle. Identifications on the boundary of this rectangle produce a round sphere, but we can easily work with two coordinates as long as we know that all pairs with latitude +90 represent the single point (North Pole). Similarly planes fly across the meridian when the longitude changes from -180 to 180.
> > >
> > > What is a practical motivation in "producing a map from the data to some Euclidean space R^n"?
> > >
> > > >definition 1.1 is vague about where I is supposed to take values
> > >
> > > The paper has no Definition 1.1. An invariant was defined in line 74: "invariant I defined as a numerical descriptor preserved by any rigid motion in R^n".
> > >
> > > >The problem that the authors are solving is in fact solved by (variants of) the Gromov-Hausdorff distance
> > >
> > > Do you agree that the Gromov-Hausdorff distance and its variants do not provide an invariant I that satisfies condition (1.1c)?
> > >
> > > Line 90: "(c) invertibility : any graph G ⊂ R^n can be reconstructed (up to rigid motion in R^n) from I(G)"
> > >
> > > >the invariant the authors introduce is basically computationally infeasible for data sets larger than maybe 100 points or so
> > >
> > > Why are 100 points relevant if there was no complete invariant at all distinguishing even graphs on 10 unordered vertices without considering 10! permutations? Such a comment can be given on any experiment: infeasible for 10x larger data.
> > >
> > > If you insist on 100 points, could you please give a link to the code that feasibly output a distance metric between rigid classes of  embedded graphs with 100 unordered vertices? Then we will certainly compare our invariants with the proposed alternative.
> > >
> > > >It is well-known that Gromov-Hausdorff can be computed for such small point clouds.
> > >
> > > We are looking forward to receiving a link to such implementations for point clouds. However, Problem 1.1 is about graphs with edges, not point clouds. Do you agree that there are exponentially many graphs with the same set of vertices (point clouds)?
> > >
> > > >the authors briefly mention metric geometry, they do not mention this nor any of the work in this direction (e.g., Greven-Pfaffelhuber-Winter).
> > >
> > > Did you mean this paper or another one? https://link.springer.com/article/10.1007/s00440-008-0169-3
> > >
> > > Yes, we can cite the proposed reference, where all metrics (similarly to the Gromov-Hausdorff metric) are defined via minimizations over infinitely many correspondences.
> > >
> > > Here is the quote from their Definition 5.1 of the Gromov-Prokhorov metric right after formula (5.9): "the infimum is taken over all isometric embeddings ϕ_X and ϕ_Y from X and Y, respectively, into some common metric space (Z, r_Z )."
> > >
> > > Do you agree that this paper does not provide any algorithms for computing this and other metrics there?
> > >
> > > > no indication is given of the scientific value of being able to characterize this data set in this fashion
> > >
> > > Lines 475-477: "the map {molecules} → {graphs on atomic centers (without chemical elements)} is injective on rigid classes. Hence this map can be theoretically inverted on its image, so we can reconstruct all chemistry from sufficiently precise geometry."
> > >
> > > >I still cannot find an answer to the question of what scientific problem has been solved
> > >
> > > Do you agree that Problem 1.1 (stated in lines 83-93) is solved?
> > >
> > > >What is the basic question here?
> > >
> > > The basic question: what is a molecular structure?
> > >
> > > Lines 478-483: "The solution to Problem 1.1 settled the long-standing challenge of properly defining a molecular structure. A traditional approach is to describe such a structure as ‘a set of unlabeled configurations that are relatively similar to each other’ ... should be rigorously defined as a class under ... rigid motion".
> > >
> > > >Why was it not handled by the approximations people tried before?
> > >
> > > Could you please give references to these approximations?
> > >
> > > Yes, we appreciate all work approximations and quoted polynomial-time cases in lines 172-174.
> > >
> > > Do you agree that Problem 1.1 requiring an exact distance metric in polynomial time for a fixed dimension is harder than computing an approximated metric?
> > >
> > > >I am willing to update my rating to a 2, as there are no ethical issues and the result is not completely trivial
> > >
> > > Could you please provide justifications for rating 2?
> > >
> > > >in the absence of more justification for the scientific value of this work I do think that is a fair description of the tone of some parts of the paper
> > >
> > > Could you please quote the parts that were meant here?

---

> > > > ### Comment · Reviewer_t58o · 2024-11-25
> > > > **Additional response**
> > > >
> > > > 1) I apologize for the confusion about the use of the term "point cloud"; I meant the finite metric space induced by the path metric on the graph.  Such metric spaces have Gromov-Hausdorff distance 0 if and only if they are isomorphic, and the analogous statement holds for restricted classes of isometries.
> > > >
> > > > 2) The reason coordinates are typically expected to take values in R^n is because in a generic metric space it is a priori very challenging to compute geodesic paths and/or centroids of sets of points.  As a consequence, it is for example very difficult to do any kind of parametric statistics.  Do you know how to efficiently compute the centroid of a collection of graphs in your metric space induced by NCD?
> > > >
> > > > 3) When you are working with sets with 10 points, computational complexity concerns do not seem to me to be very salient; even the QAP problems that arise in the generic Gromov-Hausdorff formulation can be solved by branch-and-bound methods, for instance.  Moreover, this limitation means that your method cannot be applied to many of the molecules that arise in biology, for example, most of which are substantially larger than 50 atoms.
> > > >
> > > > 4) The statement that we can "reconstruct all chemistry from sufficiently precise geometry" is too generic, and moreover I am kind of skeptical.  What does this really mean?  Can you identify a single specific question in chemistry that you can sketch an attack on using this embedding?  Again, in the context of genomics, while it is true that various people made bombastic claims that the human genome project would mean "the end of disease", there are in fact very specific clinical questions that can be posed in terms of coordinate systems on genomic data.
> > > >
> > > > 5) For finite metric spaces, none of the metrics discussed in G-P-W (or that arise in metric geometry more broadly) require considering infinitely many correspondences.

---

> > > > > ### Author Response · Authors · 2024-11-25
> > > > > **Thank you for the reply**
> > > > >
> > > > > >I apologize for the confusion about the use of the term "point cloud"; I meant the finite metric space induced by the path metric on the graph.
> > > > >
> > > > > This finite metric space can be considered a cloud (finite set) of points given by all pairwise distances because edges are not taken into account after the path metric is computed.
> > > > >
> > > > > >Such metric spaces have Gromov-Hausdorff distance 0 if and only if they are isomorphic, and the analogous statement holds for restricted classes of isometries.
> > > > >
> > > > > Do you agree that the Gromov-Hausdorff distance and isomorphism of finite metric spaces do not take into account any edges between points?
> > > > >
> > > > > The previous response underscored that "Problem 1.1 is about graphs with edges, not point clouds. Do you agree that there are exponentially many graphs with the same set of vertices (point clouds)?"
> > > > >
> > > > > >Do you know how to efficiently compute the centroid of a collection of graphs in your metric space induced by NCD?
> > > > >
> > > > > Computing a distance metric in a polynomial time is a more basic question, which was open for rigid classes of Euclidean graphs, until Problem 1.1 was solved in this paper. Complete invariants with a continuous metric are needed to unambiguously identify any rigid shape of any real molecule and recognize near-duplicates.
> > > > >
> > > > > You are right that after Problem 1.1 is solved, one can ask further questions. What is a practical motivation to compute (or even define) a centroid of graphs? In case of molecules, how would you interpret an average of a water molecule and carbon dioxide?
> > > > >
> > > > > >When you are working with sets with 10 points
> > > > >
> > > > > QM9 database has molecules up to 29 atoms.
> > > > >
> > > > > >this limitation means that your method cannot be applied to many of the molecules that arise in biology, for example, most of which are substantially larger than 50 atoms.
> > > > >
> > > > > The rebuttal mentioned that biomolecules such as DNAs or protein chains consist of atoms that are ordered along a backbone, so "all protein chains are classified by simpler invariants of ordered point clouds."  Hence invariants solving Problem 1.1 are needed for molecules with genuinely unordered atoms.
> > > > >
> > > > > >The statement that we can "reconstruct all chemistry from sufficiently precise geometry" is too generic,
> > > > >
> > > > > The previous response included the full quote:
> > > > >
> > > > > Lines 475-477: "the map {molecules} → {graphs on atomic centers (without chemical elements)} is injective on rigid classes. Hence this map can be theoretically inverted on its image, so we can reconstruct all chemistry from sufficiently precise geometry."
> > > > >
> > > > > Do you agree that the injectivity of ""the map {molecules} → {graphs on atomic centers (without chemical elements)} on rigid classes" implies that the rigid class of a molecular graph (of only atomic centers) suffices to find a unique molecule in QM9 and hence reconstruct chemical elements of all vertices?
> > > > >
> > > > > >Can you identify a single specific question in chemistry that you can sketch an attack on using this embedding?
> > > > >
> > > > > The previous response included this
> > > > >
> > > > > "The basic question: what is a molecular structure?"
> > > > >
> > > > > >Again, in the context of genomics, while it is true that various people made bombastic claims that the human genome project would mean "the end of disease", there are in fact very specific clinical questions that can be posed in terms of coordinate systems on genomic data.
> > > > >
> > > > > Our paper did not include any words "genome", "genomic", "disease", "clinical". We appreciate you interest in medical applications. Our work makes the important step towards necessary mathematical foundations by justifying the concept of a molecular structure as rigid class of embedded graphs, now efficiently identifiable due to the full solution of Problem 1.1.
> > > > >
> > > > > Because the word "bombastic" appeared again, now referring to "the human genome project" and "the end of disease", could you please confirm that our paper did not make any such claims but made the justified conclusion that the injective map {molecules} → {graphs on atomic centers} is invertible?
> > > > >
> > > > > >For finite metric spaces, none of the metrics discussed in G-P-W (or that arise in metric geometry more broadly) require considering infinitely many correspondences.
> > > > >
> > > > > Could you please quote exact results discussing how many correspondences are required?
> > > > >
> > > > > To help correctly cite this paper, the previous response asked to cite theorems about algorithms computing the Gromov-Prokhorov metric or any other metrics in polynomial time of the input size.
> > > > >
> > > > > We are waiting for responses to more questions that are left unanswered
> > > > >
> > > > > "If you insist on 100 points, could you please give a link to the code that feasibly output a distance metric between rigid classes of embedded graphs with 100 unordered vertices? Then we will certainly compare our invariants with the proposed alternative."
> > > > >
> > > > > Do you agree with reviewer RkkB? "It is very clear that this paper presents considerable work of the authors. In particular, it is clear that it is far better written and technically sound than some other papers submitted to ML conferences."

---

> > > > > > ### Comment · Reviewer_t58o · 2024-11-28
> > > > > > **One final remark**
> > > > > >
> > > > > > As a last comment about the business about reconstructing all chemistry from geometry, there is a fundamental issue which is not addressed by this paper: does the local geometry of the space in which molecular graphs are embedded reflect actual chemical properties?  This could certainly be true, but it also is by no means obvious, and has substantial bearing on whether the claims of significance for the results in this paper are meaningful.

---

> > > > > > > ### Author Response · Authors · 2024-11-28
> > > > > > > **Thank you for the great question**
> > > > > > >
> > > > > > > >does the local geometry of the space in which molecular graphs are embedded reflect actual chemical properties? This could certainly be true, but it also is by no means obvious
> > > > > > >
> > > > > > > Thank you for the great question. The paper addressed the most fundamental chemical property: the chemical composition. Because "the map {molecules} → {graphs on atomic centers (without chemical elements)} is injective on rigid classes" (lines 475-476) on all molecules (illustrated by several invariants and distance metrics in Table 3, see also Figure 4), the chemical composition can be uniquely reconstructed from the precise enough geometry of a molecular graph on atomic centers. Do you agree with this conclusion on QM9?
> > > > > > >
> > > > > > > You are right that there are many other important chemical properties, which are all determined by geometry together with atomic types (chemical elements). Since all atomic types can be inferred from geometry, other chemical properties should also be expressable in terms of pure geometry of atomic centers.
> > > > > > >
> > > > > > > Yes, finding explicit expressions of chemical properties in terms of geometric invariants is the next frontier that can be achieved only after complete invariants of Euclidean graphs are established. That is why the paper focused on justifying the concept of a molecular structure before the problem of structure-property relations can be resolved in the future.
> > > > > > >
> > > > > > > If this comment was indeed final, can we expect a fair assessment following this discussion? Thank you for reviewing the paper.

---

> ### Comment · Reviewer_RkkB · 2024-11-22
>
> Dear reviewer `t58o` and dear authors,
> I quickly read both the review as well as the response and now I am a bit sad.
>
> Firstly, I want to express that I don't think claiming that you can't find any strengths in this paper, even if you try, is very fair or nice. It is very clear that this paper presents considerable work of the authors. In particular, it is clear that it is far better written and technically sound than some other papers submitted to ML conferences. All of the other reviewers had no problems identifying at least some strengths of the paper, and I think you as well would have found something positive you could say. It is really of no sense to be deliberately mean.
> Additionally, calling the introduction "bombastic" without references to certain parts is condescending and not very productive.
> I do not criticise recommending to strongly reject this paper. I just think that this could be done in a less condescending way with a more complete assessment of the paper.
>
> On the other hand, the reply of the authors does not set standards for a civil scientific discussion either. I know that the review was not nice, but ridiculing the reviewer in response does not seem like the way to go either. (And by the way, bombastic comes from ['bombast'](https://en.wiktionary.org/wiki/bombast#English), an old word for cotton.)
>
> I am writing this both to address my sympathies for the position of the authors, as well as to say that I strongly believe in open, productive, and civil scientific discourse. If we are living in academia, we should take every step to make academia a friendly and welcoming place.
>
> Thank you and best wishes!

---

> > ### Author Response · Authors · 2024-11-22
> > **Thank you for highlighting the importance**
> >
> > Dear reviewer RkkB,
> >
> > Thank you for your kind message to all.
> >
> > >ridiculing the reviewer in response does not seem like the way to go either. (And by the way, bombastic comes from 'bombast', an old word for cotton.)
> >
> > We are not sure what exact words in our response were considered "ridiculing". Yes, the provided link to  https://en.wiktionary.org/wiki/bombast#English explains that "bombastic" refers to cotton. Sorry, we could not understand how this "cotton" is related to the paper.
> >
> > That is why we politely asked: "Could you please include the quotes that were called "bombastic"?"
> >
> > >I am writing this both to address my sympathies for the position of the authors, as well as to say that I strongly believe in open, productive, and civil scientific discourse. If we are living in academia, we should take every step to make academia a friendly and welcoming place.
> >
> > Thank you for your helpful support.

---

> > ### Comment · Reviewer_t58o · 2024-11-22
> > **Thank you**
> >
> > I'd like to thank reviewer RkkB for their comments about civility.  Reviewing my original report, although I stand by all of my criticisms of the paper, you are certainly right to observe that it was unhelpful to say that I found no strengths (the paper does seem to be more or less correct, for example) and that the blanket criticism of the tone of the exposition is both potentially hurtful and does not give anyone an opportunity to rewrite to do better.
> >
> > (Regarding the term "bombastic", this just means "giving an exaggerated sense of the importance of the work", and in the absence of more justification for the scientific value of this work I do think that is a fair description of the tone of some parts of the paper, particularly the discussion in the evaluation section.)

---

### Official Review · Reviewer_g5tG · 2024-11-10

**Soundness:** 4
**Presentation:** 3
**Contribution:** 3
**Rating:** 6
**Confidence:** 2

**Summary:**

The paper introduces a hierarchy of invariants to characterize geometric graphs, from the very simple and efficiently computable, to a complete invariant that is Lipschitz-continuous with respect to node displacement that, however, has fairly high computational complexity ( O(m^{5.5}\log^3 m) ). The paper illlustrates the invariant's ability to characterize the chemical compounds in the QM9 database.

**Strengths:**

A nice and nicely written thoretical paper. Geometric graphs arise naturally in several domains and been able to characterize them is going to be useful to several pratitioners..

**Weaknesses:**

The evaluation is a bit limited, something that can be accepted from a fundamentally theoretical paper, but what is there is difficult to understand: the authors show the minimum distance between two different chemical compounds (inter-cluster distance), but without any indication of the intra-cluster distances (maximum distance between molecules with the same chemical composition, it is hard to see how easy the clustering problem would be.

Also the claim that "the new invariants have enabled a complete classification of all molecular
graphs in the QM9 database of 130K+ (130,808) molecules given with atomic 3D coordinates.
For graphs [...] with edges, such a practical classification was previously impossible" is a bit exaggerated, and a large literature was able to solve important classification problems on QM9 even with incomplete invariant representations.

The time complexity of the complete invariant (O(m^{5.5}\log^3 m)) can be daunting for larger problems.

**Questions:**

The authors mention the time complexity, but not the space requirements of the various proposed invariants. I believe it would be very helpful to specify those as well.

---

> ### Author Response · Authors · 2024-11-17
> **Thank you for understanding the main results**
>
> >The paper introduces a hierarchy of invariants to characterize geometric graphs, from the very simple and efficiently computable, to a complete invariant that is Lipschitz-continuous with respect to node displacement
>
> Thank for correctly summarizing the main theoretical results.
>
> >has fairly high computational complexity ( O(m^{5.5}\log^3 m) ).
>
> This complexity is for the metric, the invariant NCD(G,2) in Table 2 has faster time O(m^3). More importantly, the new invariant NCD finishes the hierarchy of the faster but incomplete distance-based invariants SRV, SDV, PDD and was practically used only for a small fraction of molecular graphs, e.g. all (near) mirror images, that cannot be distinguished by the past invariants.
>
> >The paper illlustrates the invariant's ability to characterize the chemical compounds in the QM9 database.
>
> Thank you for acknowledging the practical applications.
>
> >The evaluation is a bit limited, something that can be accepted from a fundamentally theoretical paper
>
> QM9 database of 130K+ molecules with atomic coordinates seems the largest public datasets with unordered atoms, e.g. the Protein Data Bank has all atoms ordered according to a backbone, so all protein chains are classified by simpler invariants of ordered point clouds.
>
> >the authors show the minimum distance between two different chemical compounds (inter-cluster distance), but without any indication of the intra-cluster distances (maximum distance between molecules with the same chemical composition, it is hard to see how easy the clustering problem would be.
>
> Clustering by chemical composition is straightforward by comparing these compositions by strings. This and any other classification by discrete invariants only cuts the continuous space of all molecular graphs into isolated pieces.
>
> Problem 1.1 is not about clustering into disjoint classes but about a continuous parametrization of the full space of embedded graphs under rigid motion in R^n.
>
> The bottom right images in Figure 4 shows the example of (near) mirror images with a tiny EMD distance between the past invariants PDD, which are distinguished by 100 times larger distance between the new invariants NCD.
>
> >the claim that "the new invariants have enabled a complete classification of all molecular graphs in the QM9 database of 130K+ (130,808) molecules given with atomic 3D coordinates. For graphs [...] with edges, such a practical classification was previously impossible" is a bit exaggerated, and a large literature was able to solve important classification problems on QM9 even with incomplete invariant representations.
>
> To justify the claim that "a large literature was able to solve important classification problems on QM9", could you please give references to any papers that indeed solved classification problems on QM9 by complete (under rigid motion) invariants and Lipschitz continuous, all computable in polynomial time (for a fixed dimension) as in Problem 1.1?
>
> >The authors mention the time complexity, but not the space requirements of the various proposed invariants. I believe it would be very helpful to specify those as well.
>
> The SRV invariant has a linear space: m radial distances for a graph on m vertices.
>
> The SDV invariant has a quadratic space: O(m^2) pairwise distances between m vertices.
>
> The PDD invariant has a quadratic space: O(m^2) distances in the PDD matrix of maximum sizes m x (m-1).
>
> The new NCD(G,h) invariant has a space: O( m^h ( h^2+h(m-h) ) ), where the first factor O(m^h) counts the number of ordered h-vertex sequences in the vertex set of G. In fact, the completeness of NCD(G;n) for any graph G in R^n allows us to keep in memory only one Centered Representation CR(G;A) of size O(hm) for a single sequence A of h vertices.
>
> We would be happy to answer any other questions.

---

### Author Response · Authors · 2024-12-04
**Thank you for all the comments**

Dear reviewers,

thank you for all questions and justified comments, also to reviewer t58o for updating the original comment about the strengths "I am not enthusiastic about this paper and am hard-pressed to find substantial strengths" to the current version "The paper is technically sound".

We hope that that a final decision will take into account all questions and answers from the discussion and will be based on the rigorously proved results with practically important applications at the highest level of academic integrity as expected by the guides at https://www.iclr.cc/Conferences/2025/ReviewerGuide,
https://www.iclr.cc/Conferences/2025/ACGuide,
https://www.iclr.cc/Conferences/2025/SACguide.

Thank you, the authors.

---

### Meta-Review · Area_Chair_GdBy · 2024-12-20

**Metareview:**

This paper develops a new approach for producing graph invariants that can be used as features in classifying molecules, and hopefully other tasks. White the authors have produced some interesting theoretical results, showing that their invariant satisfies desirable conditions, several reviewers raised valid concerns about this work not sufficiently justifying these conditions and the main method with a scientific or experimental payoff. The authors' emphasize their belief that solving their Problem 1.1 is an interesting stand-alone theoretical contribution, but the paper would be strengthened with a better justification of this belief. A lack of sufficient comparison to alternative methods, lack of presentation clarity, and over-statement of the contributions also remain an issue with the paper. We regret that tension arose between at least one reviewer and authors during the response phase, although appreciate the patience of everyone involved in working through the miscommunication.

**Additional Comments On Reviewer Discussion:**

The rebuttal period was something of a mess. Reviewer t58o wrote what could be interpreted as an overly harsh review, which was pointed out by Reviewer RkkB, who encouraged Reviewer t58o to be more concrete with their criticism. To their credit, Reviewer t58o added significant details and remained engaged in the discussion with authors, despite its somewhat absurd nature. In particular, the authors seemed to struggle to understand much of what was written by reviewers, misinterpreted fairly standard vocabulary, and argued against valid concerns of the reviewers in a valid way. Even with no response to the reviews, this paper would likely be rejected, but the authors definitely did not help their case.

---

### Decision · Program_Chairs · 2025-01-22

Reject